# Overcoming the Incentive Collapse Paradox

**Qichuan Yin** [* 1]  **Ziwei Su** [* 1]  **Shuangning Li** [2]

## Abstract

AI-assisted task delegation is increasingly common, yet human effort in such systems is costly and typically unobserved. Recent work by Bastani & Cachon (2025); Sambasivan et al. (2021) shows that accuracy-based payment schemes suffer from incentive collapse: as AI accuracy improves, sustaining positive human effort requires unbounded payments. We study this phenomenon in a budget-constrained principal-agent framework with strategic human agents whose output accuracy depends on unobserved effort. Our first contribution is a general impossibility result showing that incentive collapse is not merely a limitation of simple linear payments, but arises for any payment rule based only on observed task accuracy. To overcome this barrier, we propose a sentinel-auditing payment mechanism that enforces a strictly positive and controllable level of human effort at finite cost, independent of AI accuracy. Building on this incentive-robust foundation, we develop an incentive-aware active statistical inference framework that jointly optimizes (i) the auditing rate and (ii) active sampling and budget allocation across tasks of varying difficulty to minimize the final statistical loss under a single budget. Experiments demonstrate improved cost-error tradeoffs relative to standard active learning and auditing-only baselines.

## 1. Introduction

AI tools are rapidly becoming embedded in everyday work and professional workflows (Krakowski et al., 2026; Madras et al., 2018; Charusaie et al., 2022; Blezek et al., 2021). In domains ranging from content moderation and medical review to auditing, legal document screening, and scientific data processing, individuals increasingly rely on AI systems to assist with complex tasks. These tools substantially improve efficiency and reduce cognitive load. A particularly common example is model-assisted data annotation, where an AI model first provides a pre-label or prediction and human annotators review, verify, or correct it through a dedicated annotation interface. Such workflows are supported by major labeling platforms (Labelbox, 2026; Label Studio, 2026) and arise in image annotation (Cormier et al., 2021), video tracking (Qiao et al., 2023), text review (Artemova et al., 2025), and other data-labeling tasks.

At the same time, human agents remain essential for handling highly complex or novel cases, exercising judgment and creativity, correcting rare but consequential errors, and providing supervision in high-stakes settings (Caro et al., 2026; De Toni et al., 2024). In an ideal world, AI systems and human agents would complement one another, combining the scalability and speed of AI with human judgment, domain knowledge, and oversight to achieve reliable and high-quality task completion.

In practice, however, ensuring effective collaboration requires careful incentive design. Human effort is costly, and the quality of task completion depends on how much attention and verification a human agent chooses to exert. Because this effort is typically unobserved, organizations must rely on performance-based compensation mechanisms to encourage sustained engagement. In many large-scale systems, compensation is tied to observed task accuracy or performance on audited items (Daniel et al., 2018). However, recent work by Bastani & Cachon (2025) formalizes what they term the **human-AI contracting paradox**: as AI accuracy improves, sustaining nonzero human effort through accuracy-based incentives becomes increasingly expensive. When AI errors are rare, a rational human agent can free-ride on the AI with minimal effort, and preventing this behavior requires payments that scale inversely with the AI error probability. As a result, accuracy-based payment schemes can collapse precisely when AI becomes most reliable.

This paper studies a general principal-agent problem arising in AI-assisted task delegation, in which task quality depends on unobserved human effort and AI suggestions are highly accurate but imperfect. Our starting point is a simple but fundamental impossibility result: if payments depend

---

[*]Equal contribution  [1]Department of Statistics, University of Chicago [2]Booth School of Business, University of Chicago. Correspondence to: Qichuan Yin <qichuan@uchicago.edu>.

*Proceedings of the 43$^{rd}$ International Conference on Machine Learning*, Seoul, South Korea. PMLR 306, 2026. Copyright 2026 by the author(s).

only on "observed accuracy," sustaining nonzero human effort becomes increasingly expensive (and indeed diverges to infinity) as AI accuracy improves. This formalizes the incentive collapse phenomenon.

Despite this negative result, we show that incentive collapse can be avoided with a simple and robust mechanism. We propose a sentinel-auditing payment scheme in which a small fraction of tasks are deliberately constructed to be difficult for the AI assistant: for example, by forcing incorrect AI outputs or by adversarially selecting tasks where the AI is unreliable. Performance on these sentinel tasks is rewarded with an explicit bonus. Crucially, the incentives induced by sentinel auditing depend on the auditing rate rather than on the AI's baseline accuracy. As a result, the mechanism enforces a strictly positive and controllable level of human effort at finite cost, even as AI accuracy approaches one. The key insight is that sentinel tasks decouple the marginal return to human effort from the AI's natural error probability, fundamentally altering the incentive landscape.

Building on this general incentive-robust framework, we study a canonical application: data collection for statistical inference, where human agents are asked to label data under a limited budget. While labeling serves as a running example, the framework applies broadly to delegated tasks whose outputs feed into downstream estimation or learning objectives. In this setting, the principal faces a dual design problem. Beyond incentivizing human effort, the principal must also decide which tasks to assign under a limited budget. Modern approaches to data-efficient inference frequently rely on active methods, which adaptively select tasks based on estimated uncertainty in order to minimize statistical error. A key implicit assumption in this literature is that once a task is selected, its quality is reliable, that is, human agents exert full and consistent effort regardless of the task.

Incentive collapse fundamentally challenges this assumption. When human agents are strategic and AI-assisted, label quality becomes endogenous: effort depends on incentives and may vary across tasks. As a result, standard active methods, which are designed under exogenous labeling cost and reliability, may allocate the labeling budget inefficiently. We address this challenge by developing an incentive-aware active statistical inference framework. In contrast to standard active learning, our approach jointly optimizes (i) the auditing rate and (ii) active sampling and budget allocation across tasks of varying difficulty under a single budget constraint. This joint design integrates incentive considerations with statistical efficiency, yielding improved cost-error tradeoffs relative to approaches that rely solely on active sampling.

Our contributions can be summarized as follows:

1. We establish an impossibility result showing that accuracy-based payment schemes suffer from incentive collapse as AI accuracy improves.

2. We propose a sentinel-auditing payment mechanism that guarantees a positive level of human effort at finite cost, independent of AI accuracy.

3. We develop an incentive-aware active statistical inference framework that jointly designs auditing and sampling policies under a single budget.

4. We provide theoretical guarantees for consistency and asymptotic normality of estimators under strategic, AI-assisted labeling.

5. We demonstrate through experiments that our approach achieves superior cost–error tradeoffs compared to standard baselines.

Taken together, our results show that effective human-AI collaboration requires explicit attention to incentive design. Improving AI accuracy alone is not sufficient; incentive-aware mechanisms are essential to ensure that increasingly capable AI systems realize their full potential in delivering reliable outcomes, including statistical inference.

## 1.1. Related Work

**Human-in-the-Loop Learning.** Human labeling remains central to modern machine learning systems, especially in ambiguous or high-stakes settings (Wu et al., 2022). Prior work improves data and cost efficiency by selecting informative tasks via active learning and adaptive sampling (Settles, 2009; Iyengar et al., 2000; Zrnic & Candès, 2024), accelerating annotation with AI assistance such as prediction sets or model suggestions (De Toni et al., 2024; Goel et al., 2023), and combining limited human-verified labels with model-generated or synthetic labels for data-efficient inference (Yin & Xin, 2026; Huang et al., 2025; Angelopoulos et al., 2023).

**Payments for Labeling and Human-AI Collaboration.** Labeling systems often use audited or "gold" items to screen agents, weight labels, and trigger bonuses or penalties (Ipeirotis et al., 2010; Oleson et al., 2011). Monetary incentives can affect participation, output, and quality (Mason & Watts, 2009). Most closely related, Bastani & Cachon (2025) show that accuracy-based incentives can become prohibitively expensive as AI accuracy improves, since agents can free-ride on near-perfect AI suggestions. To our best knowledge, this is the first work to propose an incentive-compatible mechanism that resolves this collapse and to analyze its implications for downstream procedures.

**Conflict of Interest Disclosure.** The authors declare no financial conflicts of interest.

## 2. Problem Setup

Consider a setting in which we (the principal) delegate a collection of tasks to human agents. Each task $i \in [n]$ has an observed covariate $X_i$ and has an associated (unknown) ground-truth outcome $Y_i^{\text{true}}$. For instance, in a labeling application, $X_i$ may be an input and $Y_i^{\text{true}}$ is the correct label; more generally, $Y_i^{\text{true}}$ represents the correct task output that the principal seeks to obtain. The principal assigns tasks $\{X_i\}_{i=1}^n$ to human agents, and for each $X_i$ the agent submits an output $Y_i$ intended to match $Y_i^{\text{true}}$. We assume that both the principal and the agents are rational and act to maximize their own utility.

A key feature of our setting is that human agents have access to an AI assistant, which is provided by the principal (e.g., integrated into the task interface), to improve labeling efficiency. The AI assistant is imperfect: given an input $X$, it produces an incorrect output with probability $p \in (0, 1)$, and thus produces the correct output with probability $1 - p$.

Each human agent can choose how much effort to put into completing a task. The laziest strategy is to exert zero effort and simply submit the AI assistant's output. Formally, the human agent chooses an effort level $e \in [0, 1]$, which incurs a cost $c(e)$.

Greater effort improves task quality. Specifically, conditional on the AI assistant making an error, the human agent detects and corrects the error with probability $q(e)$; if the error is not detected, the human agent accepts and uses the incorrect AI output. Therefore, the final output is incorrect only if the AI makes an error (with probability $p$) and the human agent fails to detect it (with probability $1 - q(e)$). We define output accuracy as the probability that the submitted output matches the true output. Thus, the output accuracy is $1 - p(1 - q(e))$.

From the principal's perspective, zero effort by the human agent is undesirable, as it leads to lower output accuracy. To discourage such behavior, the principal may design incentive mechanisms in the form of monetary payments. Specifically, the principal can compensate human agents according to a payment rule. Let $W_i \geq 0$ denote the (possibly random) monetary payment a human agent receives for finishing task $i$. Although $W_i$ is payment for task $i$, its value may depend on the human agent's performance on other tasks; see Section 3 for an example. Let $\mu : \mathbb{R}_+ \to \mathbb{R}$ denote the human agent's utility from monetary payments. Human agents are assumed to be rational and choose their effort level $e \in [0, 1]$ to maximize expected payoff. For a single task, this expected payoff is given by

$$\mathbb{E}\left[\mu(W_i)\right] - c(e). \tag{1}$$

where $c(e)$ is the effort cost described above.

**Assumption 2.1.** The cost function $c(\cdot)$ is increasing, dif-

ferentiable, and convex. The function $q(\cdot)$ is increasing, differentiable, and concave, and satisfies $q(1) = 1$. The utility function $\mu(\cdot)$ is strictly increasing, differentiable, and concave. We denote its inverse by $\mu^{-1}$.

The convexity of $c(\cdot)$ captures that higher effort is more costly and that the marginal cost of effort is increasing. The assumptions on $q(\cdot)$ reflect diminishing returns to effort: additional effort improves task accuracy, but at a decreasing rate, with perfect accuracy achieved at maximal effort. Finally, allowing for a general concave utility function $\mu(\cdot)$ accommodates risk aversion or diminishing marginal utility over payments; the special case $\mu(w) = w$ corresponds to risk-neutral human agents.

Recent work by Bastani & Cachon (2025) formalizes what they term the human–AI contracting paradox: under certain payment schemes, the cost required to sustain non-diminishing human effort can grow rapidly as AI accuracy improves. In Section 3, we formally establish a more general result: if payments depend only on observed task accuracy, sustaining nonzero human effort becomes increasingly expensive and indeed diverges to infinity as AI accuracy improves. In Section 4, we study how the principal can design incentives more carefully to balance cost and label accuracy to overcome such paradox. In Section 5, we show that, as a downstream application, such incentive-aware designs can be used to reduce estimation error for model parameters in active statistical inference problems.

## 3. Accuracy-Based Payments Fail at High Model Accuracy

Let us focus on a single human agent and homogeneous effort throughout this section. Suppose the human agent works with $n$ tasks. Let $Z_i \in \{0, 1\}$ indicate whether the human agent's submitted output for task $i$ is correct. Here, $Z_i \sim \text{Bern}(p_{\text{output}})$ independently, where $p_{\text{output}} = 1 - p + pq(e)$ denotes the probability that the final task output is correct, with $p$ and $q(\cdot)$ defined earlier.

We allow the payment $W_i$ allocated to task $i$ to depend on the entire vector of accuracy indicators and on additional independent randomness $\epsilon$; that is,

$$W_i = v_i(Z_1, Z_2, \ldots, Z_n, \epsilon). \tag{2}$$

This formulation is general and encompasses many commonly used quality-control mechanisms. For example, it includes per-task rewards, i.e., $v_i(Z_1, Z_2, \ldots, Z_n, \epsilon) = RZ_i$, where $R$ is a constant reward for a correct output on task $i$; it includes gold-question schemes/random spot checks, in which the principal verifies correctness on a (possibly randomized) subset of tasks and bases payment on the corresponding indicators $Z_i$, i.e., $v_i(Z_1, Z_2, \ldots, Z_n, \epsilon) = RZ_i \mathbb{1}\{Z_i \text{ is a gold question}\}$; more generally, the frame-

work allows for complex nonlinear payment rules, such as those depending on aggregate accuracy, e.g. $v_i = \frac{1}{n} R(\sum_{j=1}^{n} Z_j / n)$, where $R$ is a possibly nonlinear function of the observed accuracy level. Besides, it also includes agreement-based checks, where payment depends on the human agent's agreement with other independent human agents, since such rules can be expressed as functions of the correctness indicators $(Z_1, \ldots, Z_n)$ together with exogenous randomness capturing the behavior of other human agents. In the rest of this section, we study the agent's effort decision for a generic task. Since tasks are homogeneous, we write $W$ for the payment, dropping the subscript $i$.

Throughout this section, the human agent observes the payment rule $v_i(\cdot)$ announced by the principal, the AI error probability $p$, and the mapping $q(e)$ between effort and correction probability. Effort $e$ is chosen to maximize expected payoff in (1), taking into account the distribution of $(Z_1, \ldots, Z_n)$ induced by $p$ and $q(e)$.

We first consider linear payment schemes, in which $W$ depends linearly on $(Z_1, \ldots, Z_n)$. The linear payment scheme arises naturally in practice. For example, a human agent may receive a fixed payment for each correct output, or a fixed payment per correct output conditional on being checked (i.e., only a random subset of tasks is checked).

**Theorem 3.1** (Failure of linear accuracy-based payments). *Under Assumption 2.1, suppose the payment $W$ depends linearly on $Z_1, \ldots, Z_n$ and the utility function $\mu(\cdot)$ is risk-neutral, i.e., $\mu(w) = w$. To sustain any effort level $e > e_{\min} > 0$, the expected payment must satisfy*

$$\mathbb{E}[W] \geq C \cdot \frac{1}{p},$$

*where $C > 0$ is a constant determined by the effort $e_{\min}$.*

Theorem 3.1 shows that, under linear accuracy-based payments, sustaining any non-vanishing effort requires the expected payment to grow at least on the order of $1/p$ as the AI error probability $p$ decreases. The results of Bastani & Cachon (2025) can be viewed as a special case of Theorem 3.1. When the AI becomes highly accurate, i.e., as $p \to 0$, the principal must offer increasingly large payments to preserve incentives.

A natural question is whether more sophisticated (nonlinear) payment rules can avoid this divergence. The next result shows that, even under the most general accuracy-based payment schemes of the form (2), the expected payment must still diverge as $p \to 0$.

**Theorem 3.2** (Failure of accuracy-based payments). *Suppose the payment $W$ from the principal to the human agent is accuracy-based of the form (2), with $v$ coordinately increasing in the $Z_i$'s. Under Assumption 2.1, to sustain any*

*effort level $e > e_{\min} > 0$, the expected payment must satisfy*

$$\mathbb{E}[W] \geq \mu^{-1}\left( C \log \frac{1}{p} \right),$$

*where $C > 0$ is a constant determined by the effort $e_{\min}$. In particular, as $p \to 0$, the payment required to sustain nontrivial effort diverges.*

Theorem 3.2 establishes a general version of the *Incentive Collapse Paradox*: As AI accuracy improves (i.e., as $p$ decreases), any payment scheme based solely on task accuracy requires unbounded payments to sustain strictly positive effort. Nonlinear reward schedules do not avoid this divergence. In practice, the principal's cost may be even higher, since inducing such payments may also require additional expenditure to sample and verify output correctness.

## 4. Incentive-Robust Payments

Although the incentive collapse paradox may appear discouraging, it admits a simple and effective resolution. The key idea is to decouple the marginal benefit of human effort from the AI assistant's natural error probability $p$. To achieve this, the principal deliberately injects a small fraction of sentinel tasks, tasks on which the AI assistant is expected to fail, so that human effort is directly rewarded.

Concretely, for each task, the principal flips a coin independently. With probability $\rho \in (0, 1)$, the task is designated as a sentinel task, in which the AI assistant is deliberately forced to produce an incorrect output. With the remaining probability $1 - \rho$, the AI assistant behaves normally and produces its usual output. For sentinel tasks, the principal knows the correct output in advance, so correctness can be verified ex post. Alternatively, when such manipulation is infeasible (for example, when the AI assistant is not controlled by the principal), a similar effect can be achieved by adversarially selecting tasks that the AI is unable to output correctly and adding them to the dataset. This alternative, however, is less ideal, as it may distort the distribution of covariates $\{X_i\}$.

The principal augments the payment rule as follows. For each sentinel task that is finished correctly, the human agent receives a bonus payment $b \geq 0$. In more refined implementations, this bonus may depend on the specific covariate $x$, in which case we write $b = b(x)$. For example, the principal may assign higher bonuses to more difficult tasks, where difficulty can be estimated using the AI assistant's confidence score. Importantly, this bonus is paid only on sentinel tasks and only when the submitted output is correct.

Throughout this section, the human agent observes the announced payment rule, including the sentinel probability $\rho$, the bonus $b(\cdot)$, the AI error probability $p$, and the mapping $q(e)$ between effort and correction probability. Effort

is chosen before the realization of task types (sentinel or regular) and before correctness outcomes are realized. Thus the agent maximizes expected payoff, taking expectations over both the random designation of sentinel tasks and the submitted outcomes induced by $p$ and $q(e)$.

Intuitively, this mechanism ensures that, with probability $\rho$, exerting effort directly increases the likelihood of receiving a bonus, independently of the AI assistant's baseline accuracy. As a result, the marginal incentive for effort scales with $\rho$ rather than the AI error probability $p$, thereby neutralizing the $1/p$ effect underlying the contracting paradox. We now formalize this intuition.

**Theorem 4.1** (Effort guarantees under incentive-robust payments). *Suppose that, for each covariate $x$, the principal designates the task as a sentinel task with probability $\rho$, independently across tasks, and awards a bonus $b(x) \geq 0$ whenever a sentinel task is answered correctly. Then there exists a minimum effort level $e_{\min} > 0$ such that any rational human agent optimally chooses effort $e^*(x) \geq e_{\min}$ on every task. Moreover, the optimal effort satisfies the first-order condition*

$$\rho\mu(b(x))q'(e^*(x)) = c'(e^*(x)).$$

For illustration, consider the simple specification $q(e) = e$ and $c(e) = \frac{1}{2}e^2$. Under this model, the optimal effort admits the closed-form expression $e^*(x) = \rho\,\mu(b(x))$, which is independent of the AI accuracy. Accordingly, for a regular task, the induced correction probability on task $x$, conditional on an AI error, is $q(e^*(x)) = \rho\,\mu(b(x))$. In this case, a higher correction probability, and hence higher output accuracy, can be achieved by increasing the bonus $b(x)$, without requiring payments to diverge as the AI error probability decreases.

**Constructing sentinel tasks.** In practice, sentinel tasks can be constructed in two natural ways. First, the principal may deliberately perturb the AI-generated response before presenting it to the human agent, for example by modifying a model-generated pre-label, bounding box, or textual suggestion to introduce a controlled error. This approach is easy to scale and preserves the style of AI-assisted annotation, although care is needed to ensure that the induced errors resemble natural model mistakes. Second, the principal may use corrected historical AI errors: past instances on which the AI assistant made a mistake and human annotators corrected it. These sentinels have the advantage of preserving the natural distribution of AI failures, but require a pool of historical mistakes and are less flexible to generate on demand. An important practical question is whether annotators can distinguish such sentinel tasks from regular tasks. Under either construction, identifying a sentinel task should generally require checking the correctness of the displayed AI output itself; this is particularly difficult for corrected-history sentinels, where the displayed output is a genuine AI mistake. The two constructions can also be mixed to improve scalability while reducing detectability; see Appendix B for further discussion.

## 5. Incentive-Aware Active Statistical Inference

We now consider an application of our incentive design framework: *data labeling*. In this setting, each task consists of an input $X$ and an unobserved ground-truth label $Y$, and a human agent, assisted by an AI tool, is asked to provide a label for $X$. Importantly, data labeling is typically not an end in itself. Instead, the principal ultimately seeks accurate statistical estimation or efficient learning under a limited budget. This gives rise to a second design problem: taking the effect of effort into account, the principal needs to decide which instances to label and how to allocate payments and audits across instances with different levels of difficulty so as to minimize downstream statistical loss.

This question is closely related to the active learning literature, which studies how to adaptively select instances for labeling so as to improve downstream estimation or learning performance under a fixed or limited labeling budget. A standard modeling assumption in this literature is that, once a label is requested, its quality is reliable and independent of the sampling rule. This assumption is natural when human agents are nonstrategic or when strong quality-control mechanisms are already in place.

In our setting, however, this assumption breaks down. Under AI assistance and unobserved effort, label quality is endogenous. Human effort depends on incentives and can therefore interact with the sampling policy. In particular, when the AI is highly reliable, accuracy-based incentives provide only a weak marginal return to additional effort, giving human agents little reason to exert more effort. Consequently, inducing accurate labels on such instances can be disproportionately costly as we discussed in Section 4.

These considerations suggest that active sampling policies should be incentive-aware and explicitly incorporate mechanisms to induce effort. Building on the active statistical inference framework of Zrnic & Candès (2024), which selects instances that are most informative for statistical estimation, we develop an incentive-aware active statistical inference framework that accounts for strategic behavior. Our approach combines the incentive-robust payment mechanism developed in Section 4 with active sampling, and jointly optimizes (i) the auditing rate and (ii) active sampling and budget allocation across instances of varying difficulty under a single budget constraint. We show that this joint design leads to improved statistical efficiency and sharper error bounds relative to incentive-agnostic active inference.

## 5.1. Budget-Constrained Labeling

We now formalize the labeling process and the associated budget constraint. The principal has access to a machine learning model $f(X)$ that predicts a label $Y \in \mathbb{R}$ from $X \in \mathcal{X}$. For each instance $X_i$, the principal first decides whether to request a human label. Specifically, the principal specifies a sampling rule $\pi : \mathcal{X} \to [0, 1]$, and instance $X_i$ is selected for human labeling with probability $\pi(X_i)$. Let $\xi_i \sim \mathrm{Bern}(\pi(X_i))$ denote the indicator that $X_i$ is sampled. In standard active inference, the sampling rule is typically based on an uncertainty score $u(X_i)$, so that more informative or uncertain instances are more likely to be labeled.

Conditional on being sampled, the instance is then assigned to the incentive mechanism. As discussed earlier, because human agents may rely on AI assistance, label quality is endogenous. To induce effort, we adopt the sentinel-based payment scheme introduced in Section 4. Specifically, whenever $\xi_i = 1$, the principal independently designates the task as a sentinel task with probability $\rho \in (0, 1)$. On sentinel tasks, the AI assistant is deliberately forced to produce an incorrect label; on regular tasks, the AI assistant behaves normally. Let $\zeta_i \mid \xi_i = 1 \sim \mathrm{Bern}(1 - \rho)$ indicate whether a sampled task is regular, and define $\zeta_i \equiv 0$ if $\xi_i = 0$.

As in Section 2, annotation reliability is modeled through an effort-dependent accuracy function $q(e(X_i)) \in (0, 1]$, where $e(X_i)$ denotes the human agent's (possibly endogenous) effort on instance $X_i$. Higher effort leads to a greater probability of detecting and correcting AI errors.

Let $Y_i^{\mathrm{true}}$ denote the true label of instance $X_i$, and let $Y_i^{\mathrm{false}}$ denote the incorrect label produced by the AI assistant when it makes a mistake on instance $X_i$. Let $Y_i$ denote the label reported by the human agent. Therefore, $Y_i = Y_i^{\mathrm{true}}$ with probability $1 - p(X_i)(1 - q(e(X_i)))$, and $Y_i = f(X_i) = Y_i^{\mathrm{false}}$ with probability $p(X_i)(1 - q(e(X_i)))$.

The sampling and payment policy must satisfy an expected budget constraint. Let $b(X_i)$ denote the bonus paid when a sentinel task $X_i$ is answered correctly, let $w_0$ denote a fixed per-instance overhead payment that is independent of effort, and let $k$ denote a fixed operational cost incurred per sentinel task, for example due to adversarial construction or verification of ground truth labels. The total expected cost must be bounded by a budget $B$:

$$\mathbb{E}\left[\sum_{i=1}^{n} \Big( \rho \, b(X_i) \, q(e(X_i)) \, \xi_i + w_0 \xi_i \Big) + \rho k \right] \leq B. \quad (3)$$

Since $\xi_i \sim \mathrm{Bern}(\pi(X_i))$, this constraint is equivalent to

$$\sum_{i=1}^{n} \Big( \rho \, b(X_i) \, q(e(X_i)) + w_0 \Big) \pi(X_i) + \rho k \leq B. \quad (4)$$

Compared to standard active statistical inference, we make two key modeling distinctions. First, pointwise label quality depends on human effort and is therefore endogenously determined by the incentive scheme. Second, sampling costs are not uniform across instances. The effective cost of querying an instance depends on $\rho$ and $b(X_i)$, and must therefore be jointly optimized with the sampling policy.

## 5.2. Mean Estimation

We begin with the canonical problem of estimating the mean label $\theta^* = \mathbb{E}[Y^{\mathrm{true}}]$ under a labeling budget $B$. This setting illustrates the key principles of our approach and establishes intuition that extends to general M-estimation problems in Section 5.3.

We propose the following incentive-robust active estimator:

$$\hat{\theta}^{\pi} = \frac{1}{n} \sum_{i=1}^{n} \left[ f(X_i) + \big( Y_i - f(X_i) \big) \frac{\xi_i \, \zeta_i / (1 - \rho)}{\pi(X_i) \, q(e(X_i))} \right]. \quad (5)$$

We note that this estimator is **unbiased**. The factor $\zeta_i / (1 - \rho)$ serves as an inverse-probability correction that removes the bias induced by the sentinel mechanism. Only queried instances on which the AI behaves normally, that is, those with $(\xi_i, \zeta_i) = (1, 1)$, contribute the residual term $(Y_i - f(X_i))$. Reweighting by $1/(1 - \rho)$ corrects for the fact that such instances occur with probability $1 - \rho$ among queried points. The additional factor $1/(\pi(X_i) q(e(X_i)))$ accounts for active sampling and effort-dependent label reliability, respectively.

**Theorem 5.1.** *The incentive-robust active estimator satisfies the following central limit theorem:*

$$\sqrt{n}(\hat{\theta}^{\pi} - \theta^*) \xrightarrow{d} \mathcal{N}(0, \sigma_*^2),$$

*where* $\sigma_*^2 = \mathrm{Var}\left( f(X) + (Y - f(X)) \frac{\xi \zeta/(1-\rho)}{\pi(X) q(e(X))} \right)$. *Consequently, for any* $\hat{\sigma}^2 \xrightarrow{p} \sigma_*^2$, $C_\alpha = (\hat{\theta}^{\pi} \pm z_{1-\alpha/2} \frac{\hat{\sigma}}{\sqrt{n}})$ *is a valid* $(1 - \alpha)$-*confidence interval:*

$$\lim_{n \to \infty} P(\theta^* \in C_\alpha) = 1 - \alpha.$$

Theorem 5.1 characterizes the asymptotic distribution of the incentive-robust active estimator and shows that its accuracy is governed by the asymptotic variance $\sigma_*^2$. Consequently, to achieve the most accurate estimation under a fixed labeling budget, it is natural to seek designs that minimize $\sigma_*^2$ subject to the budget constraint (4), i.e.,

$$\min_{\pi, \rho, b} \quad \mathrm{Var}\left[ f(X) + \big( Y - f(X) \big) \frac{\xi \zeta/(1 - \rho)}{\pi(X) \, q(e(X))} \right]$$

$$\text{s.t.} \quad \sum_{i=1}^{n} \Big( \rho \, b(X_i) \, q(e(X_i)) + w_0 \Big) \pi(X_i) + \rho k \leq B.$$

This optimization problem is nontrivial because the asymptotic variance depends on several design choices simultaneously. In particular, the principal must jointly determine (i) the sampling policy $\pi(\cdot)$, which controls which instances are queried; (ii) the auditing rate $\rho$, which governs how frequently sentinel tasks are injected; and (iii) the bonus schedule $b(\cdot)$, which shapes human agent effort together with $\rho$ through incentives and hence affects label accuracy via $q(e(\cdot))$. These three components interact through both the estimator and the budget constraint, and cannot generally be optimized in isolation.

To characterize the optimal design, we first establish a variance decomposition that simplifies the optimization problem. For notational convenience, define $\tau(X_i) \triangleq \mathbb{E}\big[(Y^{\text{true}} - f(X))^2 \mid X = X_i\big]$ as the conditional prediction error.

Lemma 5.2 provides a key simplification. It shows that minimizing the asymptotic variance $\sigma_*^2$ over $(\pi, \rho, b)$ is equivalent to minimizing a weighted mean-squared error criterion, where the weights depend explicitly on the sampling probability, the auditing rate, and the effort-induced label accuracy. Building on this characterization, Examples 5.3-5.5 derive closed-form solutions under progressively more general specifications.

**Lemma 5.2.** *Minimizing*

$$\sigma_*^2 = \text{Var}\left( f(X) + (Y - f(X))\frac{\xi\zeta/(1-\rho)}{\pi(X)q(e(X))} \right)$$

*over $(\pi, \rho, b)$ is equivalent to minimizing*

$$\mathbb{E}\frac{(Y^{true} - f(X))^2}{(1-\rho)\pi(X)q(e(X))}$$

*over $(\pi, \rho, b)$.*

**Example 5.3.** Suppose $\rho$ is fixed, and $b(x) \equiv b$ is constant. By Theorem 4.1, the optimal effort level $e$ is then fixed. In this case, one can show that the variance-minimizing sampling policy satisfies $\pi(X_i) \propto \sqrt{\tau(X_i)}$.

**Example 5.4.** Suppose $b(x) \equiv b$ is constant, $q(e) = e$, $c(e) = \frac{1}{2}e^2$, and $\mu(x) = x$ (risk-neutral utility). In this case, the variance-minimizing design satisfies

$$\pi(X_i) = \frac{(B - \rho k)\sqrt{\tau(X_i)}}{\left(\rho^2 b\mu(b) + w_0\right)\sum_{i=1}^{n}\sqrt{\tau(X_i)}},$$

$$\rho = \arg\min_{\rho} \frac{\rho^2 b\mu(b) + w_0}{(1-\rho)\rho(B - \rho k)}.$$

When $k$ is negligible, we know the optimal $\rho$ is

$$\rho = \frac{-w_0 + \sqrt{w_0(w_0 + b\mu(b))}}{b\mu(b)}.$$

**Example 5.5.** Suppose $\rho$ is fixed, $q(e) = e$, $\mu(x) = x$, and $c(x) = \frac{1}{2}x^2$. The asymptotic variance $\sigma_*^2$ is minimized by the following choices of the bonus and sampling policy, $b(x_i) = \frac{\sqrt{w_0}}{\rho}$,

$$\pi(x_i) = \frac{(B - \rho k)\sqrt{\tau(x_i)}}{2w_0 \sum_{j=1}^{N}\sqrt{\tau(x_j)}}.$$

These examples reveal several key insights. First, when only some design parameters are controllable (as is often the case when market or institutional factors constrain $\rho$ or $b$), the remaining parameters optimally adjust to balance costs and variance reduction. Second, consistent with classical active learning (Zrnic & Candès, 2024), the optimal sampling policy $\pi(\cdot)$ remains proportional to $\sqrt{\tau(X_i)}$, allocating more probability to inputs with larger prediction error. Crucially, however, the sentinel design $(\rho, b)$ jointly determines the overall scale of this allocation through the budget constraint, fundamentally coupling incentive and sampling considerations.

### 5.3. General M-estimation

The mean estimation framework developed above extends naturally to general M-estimation problems, encompassing a broad class of statistical models including regression, classification, and density estimation. In standard M-estimation, the parameter $\theta$ is defined as the minimizer of the population risk based on the ground-truth label:

$$L(\theta) = \mathbb{E}\left[\ell_\theta(X, Y^{\text{true}})\right],$$

where $\ell_\theta$ is a convex loss function. Many classical estimators, including least squares and logistic regression, can be cast in this form. For notational convenience, define the per-sample losses $\ell_{\theta,i} = \ell_\theta(X_i, Y_i)$, $\ell_{\theta,i}^f = \ell_\theta(X_i, f(X_i))$, and the corresponding gradients $\nabla\ell_{\theta,i} = \nabla_\theta\ell_\theta(X_i, Y_i)$, $\nabla\ell_{\theta,i}^f = \nabla_\theta\ell_\theta(X_i, f(X_i))$.

The key structural challenge remains: labels are effort-dependent and sampling costs vary with the incentive design. Following the mean estimation approach, we adopt an active sampling policy $\pi(X_i) = \pi^{\hat\eta}(X_i) = \hat\eta u(X_i)$ proportional to an uncertainty score $u(X_i)$, where $\hat\eta$ is determined jointly with $(\rho, b)$ through the budget constraint.

The incentive-aware M-estimator is defined as

$$\hat\theta^{\hat\eta} = \arg\min_\theta L^{\hat\eta}(\theta) \tag{6}$$

where the weighted empirical loss is

$$L^{\hat\eta}(\theta) = \frac{1}{n}\sum_{i=1}^{n}\left( \ell_{\theta,i}^f + (\ell_{\theta,i} - \ell_{\theta,i}^f)\frac{\xi_i^{\hat\eta}\zeta_i/(1-\rho)}{\pi^{\hat\eta}(X_i)\,q(e(X_i))} \right),$$

where $\xi_i^{\hat\eta} \sim \text{Bern}(\pi^{\hat\eta}(X_i))$. This objective directly generalizes (5): the baseline loss $\ell_{\theta,i}^f$ uses the AI prediction,

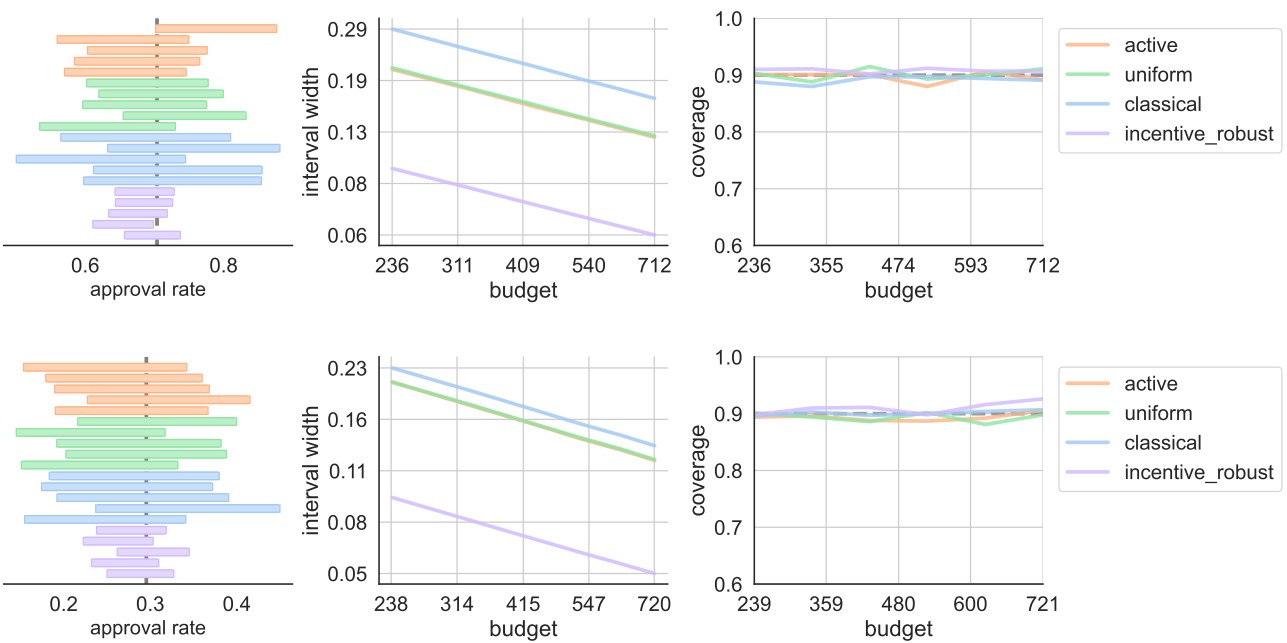

*Figure 1.* Comparison of inference performance for Biden approval (top row) and Trump approval (bottom row). The left panel displays representative confidence intervals from all methods; the middle panel shows the average interval width as a function of the budget; and the right panel reports empirical coverage, i.e., the proportion of intervals covering the true mean.

while the residual $(\ell_{\theta,i} - \ell_{\theta,i}^f)$ corrects for AI errors on non-sentinel queries using the same reweighting structure.

Under standard regularity conditions, this estimator admits asymptotically normal inference. The following theorem establishes a central limit theorem for $\hat{\theta}^{\hat{\eta}}$.

**Theorem 5.6.** *Assume the loss is smooth (Ass. A.1) and define the Hessian $H_{\theta^*} = \nabla^2 \mathbb{E}[\ell_{\theta^*}(X, Y^{true})]$. Suppose that there exists $\eta^* \in \mathcal{H}$ such that $\mathbb{P}(\hat{\eta} \neq \eta^*) \to 0$. Then, if $\hat{\theta}^{\eta^*} \xrightarrow{p} \theta^*$, we have*

$$\sqrt{n}(\hat{\theta}^{\hat{\eta}} - \theta^*) \xrightarrow{d} \mathcal{N}(0, \Sigma_*), \; where$$

$$\Sigma_* = H_{\theta^*}^{-1} \mathrm{Var}\left(\nabla \ell_{\theta^*,i}^f + \left(\nabla \ell_{\theta^*,i} - \nabla \ell_{\theta^*,i}^f\right)\Gamma_i\right) H_{\theta^*}^{-1}$$

*and $\Gamma_i = \frac{\xi_i^{\eta^*} \zeta_i / (1-\rho)}{\pi_{\eta^*}(X_i) q(e(X_i))}$. Consequently, for any $\hat{\Sigma} \xrightarrow{p} \Sigma_*$,*

$\mathcal{C}_\alpha = (\hat{\theta}_j^{\hat{\eta}} \pm z_{1-\alpha/2}\sqrt{\frac{\hat{\Sigma}_{jj}}{n}})$ *is a valid $(1-\alpha)$-confidence interval for $\theta_j^*$:*

$$\lim_{n \to \infty} \mathbb{P}(\theta_j^* \in \mathcal{C}_\alpha) = 1 - \alpha.$$

## 6. Experiments

We build on the experimental framework of Zrnic & Candès (2024) and evaluate our method on two datasets: the post-election survey data and the protein data. In both settings, a fixed budget is used to recruit human agents for labeling and conduct downstream statistical inference. We compare with

three baselines: *classical*, which uses the observed labels $Y_i$ directly; *uniform*, which uniformly samples human-labeled data and uses $f(X_i)$ for variance reduction; and *active*, which also uses $f(X_i)$ for variance reduction but adaptively queries data points following Zrnic & Candès (2024).

For all three baseline methods, payments are set to induce human effort level $e = 0.8$. Since human agents are not perfectly accurate at this effort level, the resulting estimators from these methods are biased. To enable a fair comparison, we implement an additional debiasing step for all three methods. Implementation details are provided in Appendix D. We use $q(e) = e$ and $c(e) = \frac{1}{2}e^2$ and assume risk-neutral utility $\mu(b) = b$.

### 6.1. Post-Election Survey

The Pew Research Center's post-2020 election survey dataset (Pew Research Center, 2020) contains individuals' demographic information (age, gender, etc.) as well as their approval ratings of post-election political messages from the presidential candidates. Following Zrnic & Candès (2024), we process the data so that the approval variable $Y$ is a binary response in $\{0, 1\}$. The goal is to estimate $\mathbb{E}(Y)$ using the demographic variables and to construct confidence intervals.

Assuming error-free human agents and uniform labeling costs, Zrnic & Candès (2024) show that active selection applied on top of XGBoost yields narrower confidence in-

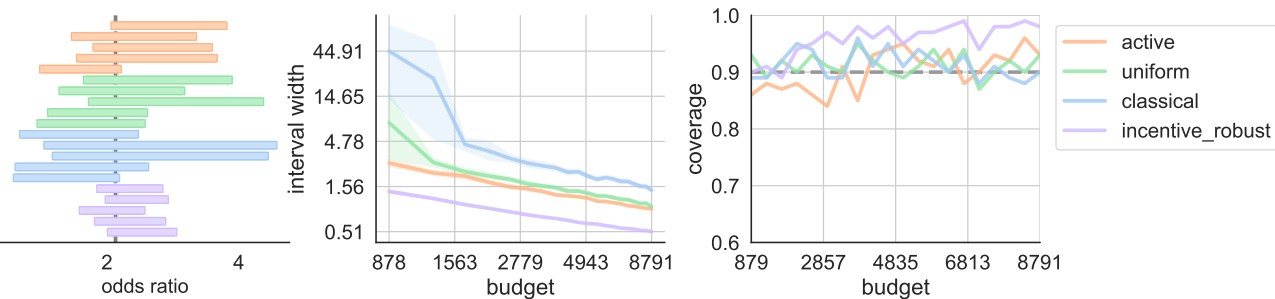

*Figure 2.* Comparison of inference performance for the odds ratio. The left panel displays representative confidence intervals from all methods; the middle panel shows the average interval width as a function of the budget; and the right panel reports empirical coverage, i.e., the proportion of intervals covering the true odds ratio.

tervals than uniform random sampling. We extend their analysis by demonstrating that even with strategic human agents, their active approach remains robust (after the debiasing step). Furthermore, our proposed method further reduces the interval length, outperforming both uniform sampling and the classical method (which relies solely on AI predictions) under the same budget constraint.

In Figure 1, we report the confidence interval width and empirical coverage of the four methods under a fixed budget: our incentive-aware active statistical inference method, the active method of Zrnic & Candès (2024), and the uniform and classical baselines. All methods achieve the nominal significance level. Our incentive-aware active sampling produces substantially shorter confidence intervals than all other methods under the same budget. Moreover, the plain active sampling method still outperforms the uniform and classical baselines.

In Figure 6, we plot the percentage of budget saved to achieve a fixed confidence interval length, compared to the three baseline methods. Our proposed incentive-aware active statistical inference framework reduces the required budget by approximately 80%.

### 6.2. Protein Data

We use the proteomics dataset from Angelopoulos et al. (2023), following Zrnic & Candès (2024). The task is to estimate the odds ratio between a protein being phosphorylated and belonging to an intrinsically disordered region (IDR). IDR is a structural property of proteins and is costly to measure experimentally, making this dataset a useful benchmark for evaluating budget-constrained inference methods.

Bludau et al. (2022) use AlphaFold (Jumper et al., 2021) to predict IDR, while Angelopoulos et al. (2023) and Zrnic & Candès (2024) improve inference using uniform and active sampling, respectively, under error-free and uniform-cost labeling assumptions.

Our experiments confirm that even with strategic human agents, the method of Zrnic & Candès (2024) continues to outperform its baselines, and our proposed approach yields further substantial improvements, as shown in Figure 2. From another perspective, Figure 7 shows that, to achieve the same confidence interval width, our method reduces the required budget by approximately 70% relative to all baselines.

## 7. Conclusion

This paper resolves the incentive collapse paradox through a sentinel-auditing mechanism that sustains positive human effort at finite cost, regardless of AI accuracy. Our theoretical analysis establishes impossibility results for accuracy-based payments and demonstrates that sentinel auditing fundamentally alters the incentive landscape by decoupling effort rewards from baseline AI error rates.

Building on this foundation, we developed an incentive-aware active statistical inference framework that jointly optimizes auditing and adaptive sampling under budget constraints. This framework explicitly accounts for the interaction between sampling policy, payment structure, and human agent behavior. Our experiments on survey and proteomics data confirm that this joint optimization yields substantial improvements over methods that treat labeling costs as uniform.

Several directions warrant further exploration. First, our framework assumes homogeneous human agents with known effort-accuracy relationships; extending to heterogeneous or learning settings presents interesting challenges. Second, while we focus on statistical estimation, the principles apply broadly to any setting where AI-assisted human decisions require quality control. Finally, the sentinel mechanism itself could be enhanced through adaptive or personalized designs that respond to individual human agent characteristics.

## Impact Statement

This work addresses incentive design for human agents in AI-assisted labeling systems. While our mechanisms improve efficiency and label quality, they involve monitoring and payment structures that affect agent compensation. Practitioners should ensure fair wages, transparent policies, and respect for agent autonomy when implementing such systems.

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

# A. Proofs

## A.1. Proof of Theorem 3.1 and Theorem 3.2

We first prove the general result Theorem 3.2.

*Proof of Theorem 3.2.* Since $W = v(Z_1, Z_2, \ldots, Z_n, \epsilon)$ and the distribution of the $Z_i$'s is fully parameterized by $p_{\text{output}}$, the expected utility of the payment, $\mathbb{E}[\mu(W)]$, can be written in the following form: $\mathbb{E}[\mu(W)] = \mu(w(p_{\text{output}}))$, for some function $w(\cdot)$. Recall that $p_{\text{output}} = 1 - p + pq(e)$ is the labeling accuracy. In particular, since $Z_1, \ldots, Z_n \sim \text{Bern}(p_{\text{output}})$, $v$ is monotone increasing in the $Z_i$'s, and $\mu$ is increasing and differentiable, the function $w(\cdot)$ is increasing and differentiable. This is because: as $p_{\text{output}}$ increases, the joint distribution of $(Z_1, \ldots, Z_n)$ shifts upward in the sense of first-order stochastic dominance. Since $v$ is coordinatewise monotone, the expectation $\mathbb{E}[v(Z_1, \ldots, Z_n, \epsilon)]$ is increasing in $p_{\text{output}}$, and because it is a finite polynomial in $p_{\text{output}}$, it is differentiable; composing with the differentiable function $\mu$ preserves differentiability.

Given effort $e$, the human agent's expected payoff is

$$\mathbb{E}[\mu(W)] - c(e) = \mu(w(p_{\text{output}})) - c(e) = \mu(w(1 - p + pq(e))) - c(e).$$

To characterize the maximizer, we differentiate with respect to $e$ and obtain

$$pq'(e)\, w'(1 - p + pq(e))\, \mu'(w(1 - p + pq(e))) - c'(e).$$

Under standard curvature conditions, this derivative is nonincreasing in $e$ and therefore can cross zero at most once. If the maximizer $e^\star$ satisfies $e^\star > e_{\min}$, then the derivative evaluated at $e_{\min}$ must be nonnegative, i.e.,

$$pq'(e_{\min})\, w'(1 - p + pq(e_{\min}))\, \mu'(w(1 - p + pq(e_{\min}))) \geq c'(e_{\min}). \tag{7}$$

Note that this needs to hold for every $p \in [0, 1]$. Thus

$$\mu'(w(x))w'(x) \geq C\frac{1}{1 - x},$$

for any $x \in [0, 1]$. Here $C$ is a constant related to $e_{\min}$. Thus

$$\mu(w(x)) \geq C\log(\frac{1}{1 - x}).$$

Thus

$$w(x) \geq \mu^{-1}(C\log(\frac{1}{1 - x})).$$

Bringing in $x = p_{\text{output}} = 1 - p + pq(e)$, we know

$$w(p_{\text{output}}) \geq \mu^{-1}(C\log(\frac{1}{p(1 - q(e))})) \geq \mu^{-1}(C\log(\frac{1}{p})).$$

Since $\mu$ is concave, we know

$$\mu(w(p_{\text{output}})) = \mathbb{E}[\mu(W)] \leq \mu(\mathbb{E}[W]),$$

and thus

$$\mathbb{E}[W] \geq w(p_{\text{output}}) \geq \mu^{-1}\left(C\log(\frac{1}{p})\right).$$

$\square$

Now let us look at the linear special case.

*Proof of Theorem 3.1.* Note that in this special case, $w$ is linear in $p_{\text{output}}$ and thus from Equation (7), we have

$$w'(1 - p + pq(e_{\min})) \geq \frac{1}{pq'(e_{\min})}c'(e_{\min}).$$

Thus

$$\mathbb{E}[W] = w(p_{\text{output}}) \geq C\frac{1}{p}.$$

$\square$

## A.2. Proof of Theorem 4.1

*Proof of Theorem 4.1.* With a minor modification, the equation from the proof of 3.2 applies here as well. We omit the details for brevity. □

## A.3. Proof of Theorem 5.1

*Proof of Theorem 5.1.* **Step 1.** We first show this estimator is unbiased. Note that

$$\mathbb{E}\left(f(X) + (Y - f(X))\frac{\xi\zeta/(1-\rho)}{\pi(X)q(e(X))}\right)$$
$$= \mathbb{E}f(X) + \mathbb{E}[(Y - f(X))\frac{1}{q(e(X))}].$$

Let event $A_1$ be AI producing wrong result and event $A_2$ be human agent correcting the wrong answer. We claim that in fact, $\mathbb{E}[Y - f(x)|X] = q(e(X))\mathbb{E}[Y^{\text{true}} - f(x)|X]$. This is because

$$\mathbb{E}[Y - f(X)|X] = \mathbb{E}[\mathbb{1}_{A_1 \cap A_2}(Y^{\text{true}} - Y^{\text{false}})|X]$$
$$= \mathbb{E}[\mathbb{1}_{A_1 \cap A_2} \mid X]\,\mathbb{E}[Y^{\text{true}} - Y^{\text{false}} \mid X]$$
$$= \mathbb{E}[\mathbb{E}[\mathbb{1}_{A_2} \mid X, A_1]\mathbb{E}[\mathbb{1}_{A_1} \mid X](Y^{\text{true}} - Y^{\text{false}})\mid X]$$
$$= q(e(X))\mathbb{E}[\mathbb{1}_{A_1}(Y^{\text{true}} - Y^{\text{false}})\mid X]$$
$$= q(e(X))\mathbb{E}[(Y^{\text{true}} - f(X))|X].$$

The last equality follows because when $A_1$ happens, we have $f(X) = Y^{\text{false}}$. Thus, $\mathbb{E}[Y - f(x)|X] = q(e(X))\mathbb{E}[Y^{\text{true}} - f(x)|X]$ and we get

$$\mathbb{E}[(Y - f(X))\frac{1}{q(e(X))}] = \mathbb{E}[Y^{\text{true}} - f(x)] = \theta^* - \mathbb{E}(f(X)).$$

Therefore this estimator is unbiased.

**Step 2.** Applying CLT and we finish the proof. □

## A.4. Proof of Lemma 5.2

*Proof of Lemma 5.2.* Minimizing the variance is equivalent to minimizing the second moment:

$$\min \mathbb{E}\left(f(X) + (Y - f(X))\frac{\xi\zeta/(1-\rho)}{\pi(X)q(e(X))}\right)^2$$
$$\Leftrightarrow \min \mathbb{E}\left(\frac{(Y - f(X))^2\xi^2\zeta^2/(1-\rho)^2}{\pi(X)^2q(e(X))^2} + 2\frac{(Y - f(X))f(X)\xi\zeta/(1-\rho)}{\pi(X)q(e(X))}\right)$$
$$\Leftrightarrow \min \mathbb{E}\left(\frac{(Y - f(X))^2}{(1-\rho)\pi(X)q(e(X))^2} + 2\frac{(Y - f(X))f(X)}{q(e(X))}\right) \quad (8)$$
$$\Leftrightarrow \min \mathbb{E}\left(\frac{(Y^{\text{true}} - f(X))^2}{(1-\rho)\pi(X)q(e(X))} + 2(Y^{\text{true}} - f(X))f(X)\right)$$
$$\Leftrightarrow \min \mathbb{E}\frac{(Y^{\text{true}} - f(X))^2}{(1-\rho)\pi(X)q(e(X))}.$$

Here we used $\mathbb{E}[\xi\zeta|X] = (1-\rho)\pi(X)$. The cross term is independent of $\pi$ and $\rho$. □

## A.5. Proof of Example 5.3 and Example 5.4

*Proof of Example 5.3.* Notice that $e$ is constant and fixed in this setting. Thus $\pi$ is the only variable in this setting. Then the optimization problem is easy to solve. □

*Proof of Example 5.4.* Notice that from Theorem 4.1, we know $e(X) \equiv \rho\mu(b)$ is constant. Thus the optimization problem is equivalent to

$$\min_{\pi,\rho} \quad \frac{1}{(1-\rho)\rho}\mathbb{E}\frac{(Y^{\text{true}} - f(X))^2}{\pi(X)}$$

$$\text{s.t.} \quad \sum_{i=1}^{n}\left(\rho^2 b\mu(b) + w_0\right)\pi(X_i) + \rho k \leq B.$$

and for fixed $\rho$, it can be shown that the optimal $\pi$ satisfying

$$\pi(X) = \lambda\sqrt{\tau(X)},$$

where

$$\tau(X_i) \triangleq \mathbb{E}\big[(Y^{\text{true}} - f(X))^2 \mid X = X_i\big].$$

Thus the optimization problem is equivalent to

$$\min_{\lambda,\rho} \quad \frac{1}{(1-\rho)\rho\lambda}$$

$$\text{s.t.} \quad \sum_{i=1}^{n}\left(\rho^2 b\mu(b) + w_0\right)\lambda\sqrt{\tau(X_i)} + \rho k \leq B.$$

Thus the optimal $\lambda$ is

$$\lambda = \frac{B - \rho k}{\left(\rho^2 b\mu(b) + w_0\right)\sum_{i=1}^{n}\sqrt{\tau(X_i)}}$$

Thus the optimization problem is equivalent to

$$\min_{\rho} \quad \frac{\rho^2 b\mu(b) + w_0}{(1-\rho)\rho(B - \rho k)}.$$

When $k$ is negligible, the optimization problem reduce to

$$\min_{\rho} \quad \frac{\rho^2 b\mu(b) + w_0}{(1-\rho)\rho}.$$

Take the derivative regard to $\rho$, we know the optimal $\rho$ is

$$\rho = \frac{-w_0 + \sqrt{w_0(w_0 + b\mu(b))}}{b\mu(b)}.$$

At this time,

$$\pi(X) = \frac{B\sqrt{\tau(X)}}{\left(\rho^2 b\mu(b) + w_0\right)\sum_{i=1}^{n}\sqrt{\tau(X_i)}}.$$

$\square$

### A.6. Proof of Example 5.5

From Theorem 4.1, we know a rational human agent would choose $e(X_i)$ satisfying

$$\rho\mu(b(X_i))q'(e(X_i)) = c'(e(X_i)).$$

Thus let $g = (c')^{-1}$, we have $e(X_i) = (c')^{-1}(\rho\mu(b(X_i))) = g(\rho\mu(b(X_i)))$. Then the question changes to

$$\min_{(\pi,\rho,b)} \mathbb{E}\frac{(Y^{\text{true}} - f(X))^2}{(1-\rho)\pi(X)g(\rho\mu(b(X)))}$$

$$\text{s.t.} \quad \sum_{i=1}^{N}\rho b(X_i)g(\rho\mu(b(X_i)))\pi(X_i) + \rho k + w_0\sum\pi(X_i) \leq B.$$

Since $\mu(x) = x$ and $c(x) = \frac{1}{2}x^2$, we know this is equivalent to

$$\min_{(\pi,\rho,b)} \frac{1}{(1-\rho)\rho} \mathbb{E} \frac{(Y^{\text{true}} - f(X))^2}{\pi(X)b(X)}$$

$$\text{s.t.} \quad \sum_{i=1}^{N} b(X_i)^2 \pi(X_i) + \frac{1}{\rho^2} w_0 \sum \pi(X_i) \leq \frac{1}{\rho^2}(B - \rho k).$$

Therefore $g(\rho\mu(b(X))) = \rho b(X)$, and the objective becomes

$$\mathbb{E}\left[\frac{(Y^{\text{true}} - f(X))^2}{(1-\rho)\,\pi(X)\,\rho b(X)}\right] = \frac{1}{(1-\rho)\rho}\mathbb{E}\left[\frac{(Y^{\text{true}} - f(X))^2}{\pi(X)b(X)}\right].$$

The constraint becomes

$$\sum_{i=1}^{N} \rho\, b(X_i) \cdot (\rho b(X_i)) \cdot \pi(X_i) + \rho k + w_0 \sum_{i=1}^{N} \pi(X_i) = \rho^2 \sum_{i=1}^{N} b(X_i)^2 \pi(X_i) + \rho k + w_0 \sum_{i=1}^{N} \pi(X_i) \leq B,$$

which is equivalent (for $\rho > 0$) to

$$\sum_{i=1}^{N} b(X_i)^2 \pi(X_i) + \frac{w_0}{\rho^2} \sum_{i=1}^{N} \pi(X_i) \leq \frac{B - \rho k}{\rho^2}.$$

Hence the problem is equivalent to

$$\min_{(\pi,\rho,b)} \frac{1}{(1-\rho)\rho}\mathbb{E}\left[\frac{(Y^{\text{true}} - f(X))^2}{\pi(X)\,b(X)}\right]$$

$$\text{s.t.} \quad \sum_{i=1}^{N} b(X_i)^2 \pi(X_i) + \frac{1}{\rho^2} w_0 \sum_{i=1}^{N} \pi(X_i) \leq \frac{1}{\rho^2}(B - \rho k).$$

Fix any $\rho \in (0, 1)$ with $B - \rho k > 0$. Then

$$\mathbb{E}\left[\frac{(Y^{\text{true}} - f(X))^2}{\pi(X)b(X)}\right] = \sum_{i=1}^{N} \frac{\tau(X_i)}{\pi(X_i)\,b(X_i)}.$$

Since the multiplicative factor $\frac{1}{(1-\rho)\rho}$ does not affect the minimizer in $(\pi, b)$ for fixed $\rho$, it suffices to solve

$$\min_{\{\pi(X_i)>0,b(X_i)>0\}_{i=1}^{N}} \sum_{i=1}^{N} \frac{\tau(X_i)}{\pi(X_i)\,b(X_i)}$$

$$\text{s.t.} \quad \rho^2 \sum_{i=1}^{N} b(X_i)^2 \pi(X_i) + w_0 \sum_{i=1}^{N} \pi(X_i) \leq B - \rho k.$$

At an optimum with $\sum_{i=1}^{N} \tau(X_i) > 0$, the constraint binds: if it were slack, one could scale $\pi(X_i) \leftarrow (1+\epsilon)\pi(X_i)$ for all $i$ (keeping $b$ fixed), which strictly decreases the objective while preserving feasibility for sufficiently small $\epsilon > 0$.

Introduce a Lagrange multiplier $\lambda \geq 0$ and consider the Lagrangian

$$\mathcal{L}(\pi, b, \lambda) = \sum_{i=1}^{N} \frac{\tau(X_i)}{\pi(X_i)b(X_i)} + \lambda\left(\rho^2 \sum_{i=1}^{N} b(X_i)^2 \pi(X_i) + w_0 \sum_{i=1}^{N} \pi(X_i) - (B - \rho k)\right).$$

Assume an interior optimum where $\pi(X_i) > 0$ and $b(X_i) > 0$ for all $i$ with $\tau(X_i) > 0$. The first-order conditions for each $i$ are

$$\frac{\partial \mathcal{L}}{\partial \pi(X_i)} = -\frac{\tau(X_i)}{\pi(X_i)^2 b(X_i)} + \lambda\left(\rho^2 b(X_i)^2 + w_0\right) = 0,$$

and

$$\frac{\partial \mathcal{L}}{\partial b(X_i)} = -\frac{\tau(X_i)}{\pi(X_i)\,b(X_i)^2} + \lambda\left(2\rho^2 b(X_i)\pi(X_i)\right) = 0.$$

From the first condition,

$$\frac{\tau(X_i)}{\pi(X_i)^2 b(X_i)} = \lambda\left(\rho^2 b(X_i)^2 + w_0\right). \tag{9}$$

From the second condition,

$$\frac{\tau(X_i)}{\pi(X_i)\,b(X_i)^2} = 2\lambda\rho^2 b(X_i)\pi(X_i). \tag{10}$$

Multiplying (10) by $\pi(X_i)$ yields

$$\frac{\tau(X_i)}{b(X_i)^2} = 2\lambda\rho^2 b(X_i)\pi(X_i)^2, \qquad \text{so} \qquad \pi(X_i)^2 = \frac{\tau(X_i)}{2\lambda\rho^2 b(X_i)^3}. \tag{11}$$

Similarly, rearranging (9) gives

$$\pi(X_i)^2 = \frac{\tau(X_i)}{\lambda b(X_i)\left(\rho^2 b(X_i)^2 + w_0\right)}. \tag{12}$$

Equating (11) and (12) and canceling the common factor $\tau(X_i)/(\lambda b(X_i)) > 0$, we obtain

$$\frac{1}{2\rho^2 b(X_i)^2} = \frac{1}{\rho^2 b(X_i)^2 + w_0},$$

which implies

$$\rho^2 b(X_i)^2 + w_0 = 2\rho^2 b(X_i)^2 \quad \Longrightarrow \quad \rho^2 b(X_i)^2 = w_0 \quad \Longrightarrow \quad b(X_i) = \frac{\sqrt{w_0}}{\rho}.$$

Hence the optimal $b(\cdot)$ is constant over $\{X_1, \ldots, X_N\}$:

$$b^\star(X_i) = \frac{\sqrt{w_0}}{\rho} \qquad \forall i.$$

Substituting $b^\star(X_i) = \frac{\sqrt{w_0}}{\rho}$ into (9), note that $\rho^2 b^\star(X_i)^2 = w_0$ and thus $\rho^2 b^\star(X_i)^2 + w_0 = 2w_0$. Therefore

$$\frac{\tau(X_i)}{\pi(X_i)^2\,b^\star(X_i)} = \lambda \cdot 2w_0, \qquad \text{so} \qquad \pi(X_i) = \sqrt{\frac{\tau(X_i)}{2\lambda w_0\,b^\star(X_i)}}.$$

Because $b^\star(X_i)$ is constant in $i$, this shows

$$\pi^\star(X_i) \propto \sqrt{\tau(X_i)}.$$

To determine the proportionality constant, use that the budget constraint binds at the optimum:

$$\rho^2 \sum_{i=1}^{N} b^\star(X_i)^2 \pi^\star(X_i) + w_0 \sum_{i=1}^{N} \pi^\star(X_i) = B - \rho k.$$

Since $\rho^2 b^\star(X_i)^2 = w_0$, the left-hand side equals

$$w_0 \sum_{i=1}^{N} \pi^\star(X_i) + w_0 \sum_{i=1}^{N} \pi^\star(X_i) = 2w_0 \sum_{i=1}^{N} \pi^\star(X_i),$$

hence

$$\sum_{i=1}^{N} \pi^\star(X_i) = \frac{B - \rho k}{2w_0}.$$

Let $r_i \triangleq \sqrt{\tau(X_i)}$ and $R \triangleq \sum_{j=1}^{N} r_j$. Then the unique proportional allocation satisfying the above sum constraint is

$$\pi^\star(X_i) = \frac{B - \rho k}{2w_0} \cdot \frac{r_i}{R} = \frac{B - \rho k}{2w_0} \cdot \frac{\sqrt{\tau(X_i)}}{\sum_{j=1}^{N} \sqrt{\tau(X_j)}}.$$

This completes the derivation of the optimal $(\pi, b)$ for fixed $\rho$ under the stated interior conditions.

### A.7. Proof of Theorem 5.6

*Proof.* We follow the similar proof process of Theorem 1 in (Zrnic & Candès, 2024). We first formally stating the required smoothness assumption.

**Assumption A.1** (Smoothness). The loss $\ell$ is smooth if:

- $\ell_\theta(x, y)$ is differentiable at $\theta^*$ for all $(x, y)$;

- $\ell_\theta$ is locally Lipschitz around $\theta^*$: there is a neighborhood of $\theta^*$ such that $\ell_\theta(x, y)$ is $C(x, y)$-Lipschitz and $\ell_\theta(x, f(x))$ is $C(x)$-Lipschitz in $\theta$, where $\mathbb{E}[C(X, Y^{\text{true}})^2] < \infty, \mathbb{E}[C(X)^2] < \infty$;

- $L(\theta) = \mathbb{E}[\ell_\theta(X, Y^{\text{true}})]$ and $L^f(\theta) = \mathbb{E}[\ell_\theta(X, f(X))]$ have Hessians, and $H_{\theta^*} = \nabla^2 L(\theta^*) \succ 0$.

let $L_{\theta,i}^\eta = \ell_\theta(X_i, f(X_i)) + (\ell_\theta(X_i, Y_i) - \ell_\theta(X_i, f(X_i))) \frac{\xi_i^\eta \zeta_i/(1-\rho)}{\pi_\eta(X_i)q(e(X_i))}$, where $\pi_\eta(X_i) = \eta u(X_i)$ and $\xi_i^\eta \sim B(\pi_\eta(X_i))$. We define $\nabla L_{\theta,i}^\eta$ analogously, replacing the losses with their gradients. Given a function $g$, let

$$\mathbb{G}_n[g(L_\theta^\eta)] := \frac{1}{\sqrt{n}} \sum_{i=1}^n \left( g(L_{\theta,i}^\eta) - \mathbb{E}[g(L_{\theta,i}^\eta)] \right); \quad \mathbb{E}_n[g(L_\theta^\eta)] := \frac{1}{n} \sum_{i=1}^n g(L_{\theta,i}^\eta).$$

We similarly use $\mathbb{G}_n[g(\nabla L_\theta^\eta)], \mathbb{E}_n[g(\nabla L_\theta^\eta)]$, etc. Notice that $\mathbb{E}_n[L_\theta^{\hat\eta}] = L^{\pi_{\hat\eta}}(\theta)$.

By the differentiability and local Lipschitzness of the loss, for any $h_n = O_P(1)$ we have

$$\mathbb{G}_n[\sqrt{n}(L_{\theta^*+h_n/\sqrt{n}}^{\eta^*} - L_{\theta^*}^{\eta^*}) - h_n^\top \nabla L_{\theta^*}^{\eta^*}] \xrightarrow{p} 0.$$

By definition, this is equivalent to

$$n\mathbb{E}_n[L_{\theta^*+h_n/\sqrt{n}}^{\eta^*} - L_{\theta^*}^{\eta^*}] = n(L(\theta^* + h_n/\sqrt{n}) - L(\theta^*)) + h_n^\top \mathbb{G}_n[\nabla L_{\theta^*}^{\eta^*}] + o_P(1),$$

where $L(\theta) = \mathbb{E}[\ell_\theta(X, Y^{\text{true}})]$ is the population loss. A second-order Taylor expansion now implies

$$n\mathbb{E}_n[L_{\theta^*+h_n/\sqrt{n}}^{\eta^*} - L_{\theta^*}^{\eta^*}] = \frac{1}{2}h_n^\top H_{\theta^*} h_n + h_n^\top \mathbb{G}_n[\nabla L_{\theta^*}^{\eta^*}] + o_P(1).$$

At the same time, since $\mathbb{P}(\hat\eta \neq \eta^*) \to 0$, we have

$$n\mathbb{E}_n[L_{\theta^*+h_n/\sqrt{n}}^{\hat\eta} - L_{\theta^*}^{\hat\eta}] = n\mathbb{E}_n[L_{\theta^*+h_n/\sqrt{n}}^{\eta^*} - L_{\theta^*}^{\eta^*}] + o_P(1).$$

Putting everything together, we have shown

$$n\mathbb{E}_n[L_{\theta^*+h_n/\sqrt{n}}^{\hat\eta} - L_{\theta^*}^{\hat\eta}] = \frac{1}{2}h_n^\top H_{\theta^*} h_n + h_n^\top \mathbb{G}_n[\nabla L_{\theta^*}^{\eta^*}] + o_P(1).$$

We apply the previous display with $h_n = \hat h_n := \sqrt{n}(\hat\theta^{\hat\eta} - \theta^*)$ (which is $O_P(1)$ by the consistency of $\hat\theta^{\eta^*}$; $h_n = \tilde h_n := -H_{\theta^*}^{-1}\mathbb{G}_n[\nabla L_{\theta^*}^{\eta^*}]$:

$$n\mathbb{E}_n[L_{\hat\theta^{\hat\eta}}^{\hat\eta} - L_{\theta^*}^{\hat\eta}] = \frac{1}{2}\hat h_n^\top H_{\theta^*} \hat h_n + \hat h_n^\top \mathbb{G}_n[\nabla L_{\theta^*}^{\eta^*}] + o_P(1);$$

$$n\mathbb{E}_n[L_{\theta^*+\tilde h_n/\sqrt{n}}^{\hat\eta} - L_{\theta^*}^{\hat\eta}] = \frac{1}{2}\tilde h_n^\top H_{\theta^*} \tilde h_n + \tilde h_n^\top \mathbb{G}_n[\nabla L_{\theta^*}^{\eta^*}] + o_P(1).$$

By the definition of $\hat\theta^{\hat\eta}$, the left-hand side of the first equation is smaller than the left-hand side of the second equation. Therefore, the same must be true of the right-hand sides of the equations. If we take the difference between the equations and complete the square, we get

$$\frac{1}{2}\left(\sqrt{n}(\hat\theta^{\hat\eta} - \theta^*) - \tilde h_n\right)^\top H_{\theta^*}\left(\sqrt{n}(\hat\theta^{\hat\eta} - \theta^*) - \tilde h_n\right) + o_P(1) \leq 0.$$

Since the Hessian $H_{\theta^*}$ is positive-definite, it must be the case that $\sqrt{n}(\hat{\theta}^{\hat{\eta}} - \theta^*) - \tilde{h}_n \xrightarrow{p} 0$. By the central limit theorem, $\tilde{h}_n = -H_{\theta^*}^{-1}\mathbb{G}_n[\nabla L_{\theta^*}^{\eta^*}]$ converges to $\mathcal{N}(0, \Sigma_*)$ in distribution, where

$$\Sigma_* = H_{\theta^*}^{-1}\mathrm{Var}\left(\nabla\ell_{\theta^*}(X, f(X)) + (\nabla\ell_{\theta^*}(X, Y) - \nabla\ell_{\theta^*}(X, f(X)))\frac{\xi^{\eta^*}\zeta/(1-\rho)}{\pi_{\eta^*}(X)q(e(X))}\right)H_{\theta^*}^{-1}.$$

The final statement thus follows by Slutsky's theorem.

$\square$

## B. Identifiability of Sentinel Tasks

A practical concern is whether strategic annotators can identify sentinel tasks and exert effort only on those tasks. In the main analysis, we focus on the idealized case where the sentinel status is not directly observable. In this section, we discuss when this assumption is plausible, how the mechanism extends when annotators can form imperfect posterior beliefs, and when identifiability becomes a limitation.

The identifiability of sentinel tasks depends on how they are constructed. The most favorable case is the corrected-history construction, where sentinel tasks are drawn from past instances on which the AI system made genuine mistakes. In this case, the AI output shown to the annotator is a real historical AI error. Therefore, there is no special pattern that distinguishes sentinel tasks from ordinary AI-assisted tasks.

A second construction deliberately modifies the AI suggestion before presenting it to the annotator. This construction is more scalable, but it requires more care. When the modification preserves the style of the AI output and produces errors that are plausible under the AI's natural error distribution, identifying sentinel status remains difficult.

If the induced errors are too different from the natural AI error distribution, annotators may learn to recognize them. However, this concern is less severe in settings where the cost of checking the AI output is much larger than the cost of correcting an identified error. For example, in proofreading or polishing AI-generated text, the main effort often lies in carefully reading the text and identifying subtle factual, grammatical, or stylistic issues; once such an issue is found, making the correction is usually relatively easy. In such settings, the main role of sentinel auditing is to incentivize annotators to verify the AI output carefully, rather than to perfectly match the natural distribution of AI errors. Once an error is detected, correcting it is relatively inexpensive. Therefore, even if sentinel errors are somewhat easier to correct than natural AI errors, they can still serve their incentive purpose as long as detecting them requires the same substantive verification effort.

For continuous-valued labels, this creates a margin-of-error tradeoff: if the modification is too small, a high-effort worker may reasonably regard the original AI answer as acceptable; if the modification is too large, the sentinel may become obvious. Thus, for real-valued tasks, sentinel construction should respect the task-specific tolerance level and produce errors that are detectable under careful verification but not obvious without effort.

We next discuss the intermediate case where annotators can form imperfect beliefs about whether a task is sentinel. As long as they cannot perfectly identify sentinel tasks (i.e., their belief is strictly less than 1), the incentive effect can still be sustained. Suppose that for a non-sentinel task, the annotator believes it is sentinel with probability $\rho_{\mathrm{posterior}}(X)$. Then, similarly to Theorem 4.1, the first-order condition becomes

$$\rho_{\mathrm{posterior}}(X)\mu(b(X))q'(e(X)) = c'(e(X)),$$

so a feasible effort level $e$ for non-sentinel tasks can still be supported.

Therefore, imperfect identifiability does not eliminate incentives. It changes the effective auditing probability from the design parameter $\rho$ to the worker's posterior belief $\rho_{\mathrm{posterior}}$. As long as this posterior belief is not zero on ordinary tasks, a positive effort level can still be supported.

The mechanism fails only in the extreme case where annotators can perfectly identify sentinel tasks before exerting effort. If $\rho_{\mathrm{posterior}} = 0$ for non-sentinel tasks, then the worker has no incentive to exert effort on non-sentinel tasks.

To further examine this extension, we add a sensitivity analysis in which the annotator's posterior belief is set to $\rho_{\mathrm{posterior}} = 0.8\rho$. All other experimental parameters are kept the same as in the main experiments. The results are shown in Figure 3. As expected, the performance is somewhat worse than in the case where sentinel tasks are fully indistinguishable ($\rho_{\mathrm{posterior}} = \rho$), but our method still outperforms the baselines.

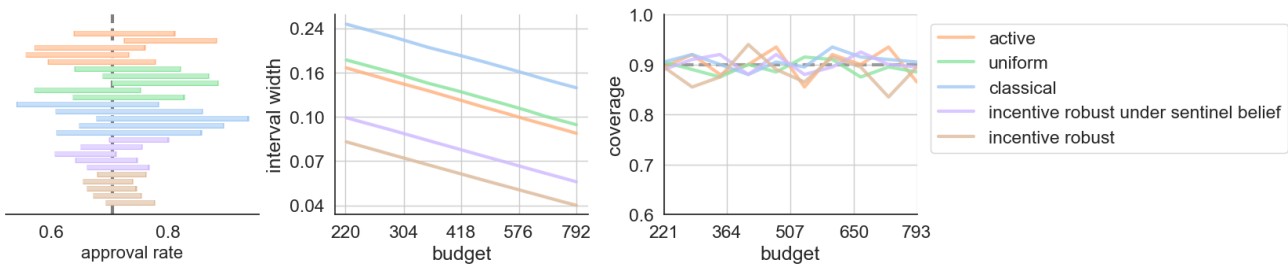

*Figure 3.* Comparison of inference performance for Biden approval with posterior belief. The left panel displays representative confidence intervals from all methods; the middle panel shows the average interval width as a function of the budget; and the right panel reports empirical coverage, i.e., the proportion of intervals covering the true mean.

## C. Extension to Heterogeneous Task Difficulty

The main text focuses on a homogeneous task setting for clarity. In practice, however, tasks may differ in difficulty, and the cost of verifying labels or constructing sentinel tasks may also vary across tasks. Our framework can be extended to this setting in a straightforward way.

Suppose each task $X_i$ has a difficulty level $h(X_i) \in \{1, \ldots, H\}$. We allow the effort–accuracy function, auditing probability, bonus, and sentinel-construction cost to depend on difficulty:

$$q_i(e) = q(e(X_i), h(X_i)), \qquad \rho_i = \rho(h(X_i)), \qquad b_i = b(X_i, h(X_i)), \qquad k_i = k(h(X_i)).$$

Then the incentive condition in Theorem 4.1 still holds accordingly. Thus, harder tasks can be assigned larger bonuses or higher auditing probabilities in order to induce the desired effort level.

At the same time, the cost of obtaining ground truth or constructing sentinel tasks can also depend on difficulty, represented by $k(h(X_i))$. This changes the budget constraint to

$$\mathbb{E}\left[ \sum_{i=1}^{n} \Big( \rho(h(X_i)) b(X_i, h(X_i)) q(e(X_i), h(X_i)) \xi_i + w_0 \xi_i + \rho(h(X_i)) k(h(X_i)) \Big) \right] \leq B.$$

To solve the above optimization problem, one possible approach is to decompose it into $H$ subproblems, each corresponding to a difficulty level. The budget allocation then proceeds in two stages: in the first stage, the total budget is allocated across difficulty levels; in the second stage, we apply the results from the current version of the paper (for homogeneous difficulty) to optimize the mechanism within each difficulty class. We leave a full characterization of the optimal heterogeneous policy to future work.

## D. Experiment Details

Codes of our experiments can be found at
https://github.com/su-z/overcoming-the-incentive-collapse-paradox.

This section details the experimental setup (for both experiments), the simulation of the labeling process, and the specific functional forms used for the economic components of the model. We adapt the experimental setup of Zrnic & Candès (2024) to our settings, introducing new methods.

For models predicting binary outcomes $Y^{\text{true}} \in \{0, 1\}$, our model might predict a real number between 0 and 1. Denote this number by $\hat{p}(X)$, which we treat as the model's probability to predict class 1. The probability of the model making a correct prediction given the true label $Y^{\text{true}}$ is given by the Bernoulli likelihood:

$$P(\text{correct} \mid \hat{p}, Y^{\text{true}}) = \hat{p} \cdot Y^{\text{true}} + (1 - \hat{p}) \cdot (1 - Y^{\text{true}}). \tag{13}$$

Conversely, the probability of the AI making an error on an instance is:

$$p(X) = 1 - P(\text{correct} \mid \hat{p}, Y^{\text{true}}) = 1 - (\hat{p} \cdot Y^{\text{true}} + (1 - \hat{p}) \cdot (1 - Y^{\text{true}})). \tag{14}$$

---

**Algorithm 1** Incentive-Robust Active Statistical Inference

---

**Require:** Unlabeled data $\{(X_i, f(X_i))\}_{i=1}^{n}$, budget $B$, sentinel rate $\rho \in (0, 1)$, cost parameters $w_0, k$, error level $\alpha$
**Ensure:** Estimate $\hat{\theta}^\pi$ and confidence interval

1: **Step 1: Design incentive mechanism**
2: Compute per-instance uncertainty measure $\ell(x_i)$
3: Set bonus $b = \frac{\sqrt{w_0}}{\rho}$ and induced effort $e = \rho b$
4: Compute expected cost per sample: $c_S = w_0 + \rho k$
5: **Step 2: Allocate sampling budget**
6: Compute unscaled weights: $\tilde{\pi}_i = \sqrt{\ell(x_i)}$
7: Compute expected total cost: $C_{\exp} = \sum_{i=1}^{n} \tilde{\pi}_i c_S$
8: Compute scaling factor: $\lambda = B/C_{\exp}$
9: Set sampling probabilities: $\pi(x_i) = \min(1, \lambda \tilde{\pi}_i)$
10: **Step 3: Sample and query human agents**
11: **for** $i = 1$ to $n$ **do**
12:     Draw sampling indicator: $\xi_i \sim \text{Bernoulli}(\pi(x_i))$
13:     **if** $\xi_i = 1$ **then**
14:         Draw sentinel indicator: $\zeta_i \sim \text{Bernoulli}(1 - \rho)$
15:         **if** $\zeta_i = 0$ **then**
16:             Create sentinel task (force AI error)
17:         **end if**
18:         human agent exerts effort $e$ and submits label $Y_i$
19:         Labeling accuracy: $1 - p_i(1 - q(e))$
20:         **if** $\zeta_i = 0$ and $Y_i$ is correct **then**
21:             Pay bonus $b$
22:         **end if**
23:         Pay base payment $w_0$
24:     **end if**
25: **end for**
26: **Step 4: Compute corrected estimator**
27: Estimate parameter via equation (5).
28: **Step 5: Construct confidence interval**
29: Compute asymptotic variance $\hat{\Sigma}$ via Theorem 5.6
30: **return** $\hat{\theta}^\pi$ and CI: $\left( \hat{\theta}_j^\pi \pm z_{1-\alpha/2} \sqrt{\frac{\hat{\Sigma}_{jj}}{n}} \right)$

---

We simulate imperfect error detection by human agents. A human agent with effort level $e$ detects and corrects an AI error with probability $q(e)$. The label generation process for each instance is as follows:

1. **AI Prediction:** The model outputs a probability $\hat{p}$ and makes a prediction $f(X) = \mathbb{I}(\hat{p} > 0.5)$.

2. **Check for AI Error:** Determine whether the AI prediction is incorrect $\mathbb{I}(f(X) \neq Y^{\text{true}})$.

3. **Human Agent Correction:** If an error occurred ($\mathbb{I}(f(X) \neq Y^{\text{true}}) = 1$), the human agent detects and corrects it with probability $q(e)$. Sample a detection indicator $a \sim \text{Bernoulli}(q(e))$.

4. **Final Label:**

   - If $\mathbb{I}(f(X) \neq Y^{\text{true}}) = 1$ and $a = 1$ (error detected and corrected), the human agent outputs the correct label: $Y = Y^{\text{true}}$.
   - Otherwise (no error, or error not detected), the human agent accepts the AI prediction: $Y = f(X)$.

This process yields an empirical labeling accuracy of $P(Y \text{ correct}) = 1 - p(1 - q(e))$, where $p$ is the AI error rate.

In our experiments, we take $q(e) = e, c(e) = \frac{1}{2}e^2$, and risk-neutral utility function $\mu(b) = b$.

**Baseline Methods.** Zrnic & Candès (2024) introduced the parameter $\tau \in [0, 1]$:

$$\pi(x_i) = (1 - \tau)\pi_{\text{active}}^*(x_i) + \tau \cdot \frac{B}{n}, \tag{15}$$

where $\tau = 0$ corresponds to pure active sampling and $\tau = 1$ corresponds to uniform sampling, and tune $\tau$ to minimize variance in the post-election survey data experiment and hard code $\tau = 0.5$ for the protein property experiment. Our incentive-robust method does not use $\tau$ but for comparison we still include $\tau$ for baselines.

The estimator given in Zrnic & Candès (2024) has the following form:

$$\hat{\theta} = \frac{1}{n} \sum_{i=1}^{n} \left( f(X_i) + (Y_i - f(X_i)) \frac{\xi_i}{\pi(X_i)} \right).$$

But this is only unbiased for labels that are completely accurate. In our experiments the label has errors, which introduce a bias. To correct this bias, for our baseline, we actually use the following estimator (in our case $e(X_i)$ is constant)

$$\hat{\theta}_{\text{base}} = \frac{1}{n} \sum_{i=1}^{n} \left( f(X_i) + (Y_i - f(X_i)) \frac{\xi_i}{\pi(X_i)q(e(X_i))} \right).$$

For the uniform estimator, we simply set $\pi(X_i) \equiv \pi_{\text{unif}}$:

$$\hat{\theta}_{\text{unif}} = \frac{1}{n} \sum_{i=1}^{n} \left( f(X_i) + (Y_i - f(X_i)) \frac{\xi_i}{\pi_{\text{unif}}q(e(X_i))} \right).$$

For the classical estimator, we assume a symmetric noise model (since we do not use $f$ to identify AI errors) and apply the standard correction:

$$\hat{\theta}_{\text{class}} = \frac{1}{n} \sum_{i=1}^{n} \frac{\xi_i}{\pi_{\text{unif}}} \frac{Y_i + q(e(X_i)) - 1}{2q(e(X_i)) - 1}.$$

### D.1. Robustness to Utility and Cost Misspecification

Our theoretical analysis assumes that the principal has a correctly specified model of the annotator's utility and cost functions. This assumption is useful for deriving the optimal bonus and auditing design, but it may be strong in practice. Annotator behavior can vary across workers, tasks, and platforms, and the true effort cost may not be known exactly.

In practical deployments, the principal can use historical annotation data, pilot studies, or platform-level behavioral data to estimate the relevant behavioral parameters. These estimates can also be updated adaptively over time.

Importantly, the sentinel-auditing mechanism is not all-or-nothing with respect to model specification. Even when the utility and cost functions are misspecified, the mechanism can still create a direct incentive for annotators to verify the AI output. Misspecification may lead to a suboptimal choice of the bonus or auditing rate, but our method still guarantees non-vanishing human effort and outperforms baselines that ignore strategic behavior.

**Experimental setup.** The experimental setup follows the main experiment, including the same Pew ATP Wave 79 data, outcome preprocessing, covariates, prediction model, baseline methods, and evaluation metrics. In the main experiment, the principal designs the mechanism under the behavioral model $\mu(b) = b$, $q(e) = e$, and $c(e) = e^2/2$, so the induced effort is predicted by the first-order condition $\rho b = c'(e) = e$. In this sensitivity experiment, the principal still optimizes payments and sampling probabilities under this assumed model, but the simulated annotator follows a weaker true response: in each Monte Carlo trial, we draw $k \sim \mathrm{Uniform}(0.25, 0.75)$ and set the actual utility function to $\mu_{\mathrm{actual}}(b) = kb$ and thus effort to $k$ times the effort predicted by the assumed model. We report 90% confidence intervals, average interval widths, and empirical coverage. Figure 4 reports a sensitivity analysis in which the mechanism is optimized under misspecified utility

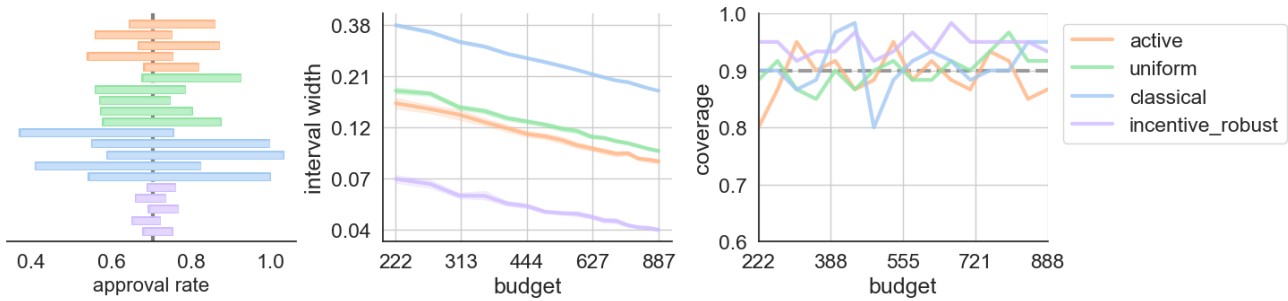

*Figure 4.* Comparison of inference performance for Biden approval. The left panel displays representative confidence intervals from all methods; the middle panel shows the average interval width as a function of the budget; and the right panel reports empirical coverage, i.e., the proportion of intervals covering the true mean.

and cost functions. The results suggest that our method continues to induce non-vanishing human effort and yields tighter confidence intervals than baselines that ignore strategic behavior. This finding supports the main message of the paper: explicitly accounting for incentives, even with an imperfect behavioral model, is preferable to treating human labels as exogenous.

## D.2. Experiment with Continuous Outcomes

Although our main experiments focus on binary outcomes, our theoretical framework applies to continuous outcomes. To demonstrate this, we conduct an additional experiment using the 2019 ACS PUMS data for California (Ding et al., 2021), where the outcome $Y$ is individual income. The ACS PUMS is a large-scale annual survey conducted by the U.S. Census Bureau and includes demographic variables such as age, sex, education, and income.

The target parameter is the coefficient of age in a linear regression of income on age and sex. This gives a standard continuous-outcome M-estimation problem, where the loss is the squared loss and the target parameter is defined by the population least-squares objective. We use the same experimental setup as in the main experiments: an AI prediction model first provides baseline predictions, and the principal allocates a fixed budget across sampling and sentinel auditing. We compare our incentive-aware method with the same baselines as before. For each budget level, we construct confidence intervals for the age coefficient and evaluate both interval width and empirical coverage across repeated Monte Carlo trials. As shown in Figure 5, our method yields tighter confidence intervals than baselines that ignore strategic behavior, supporting the applicability of our approach beyond discrete labeling tasks.

## D.3. More Experiment Results

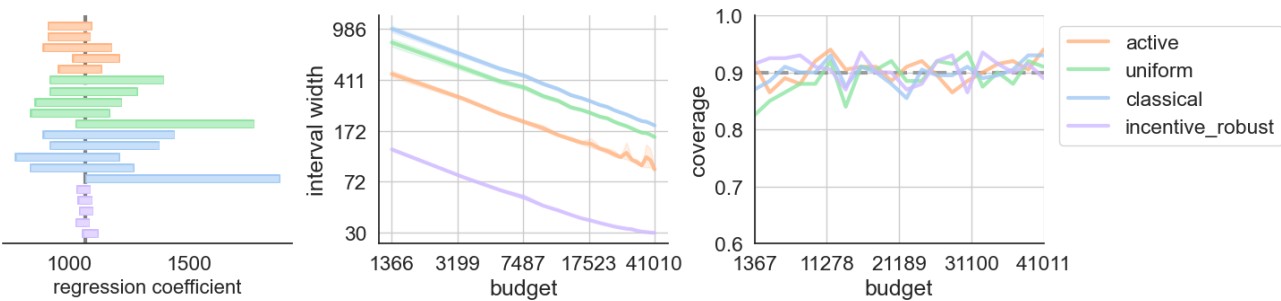

*Figure 5.* Comparison of inference performance for ACS PUMS data. The left panel displays representative confidence intervals from all methods; the middle panel shows the average interval width as a function of the budget; and the right panel reports empirical coverage, i.e., the proportion of intervals covering the true age coefficient.

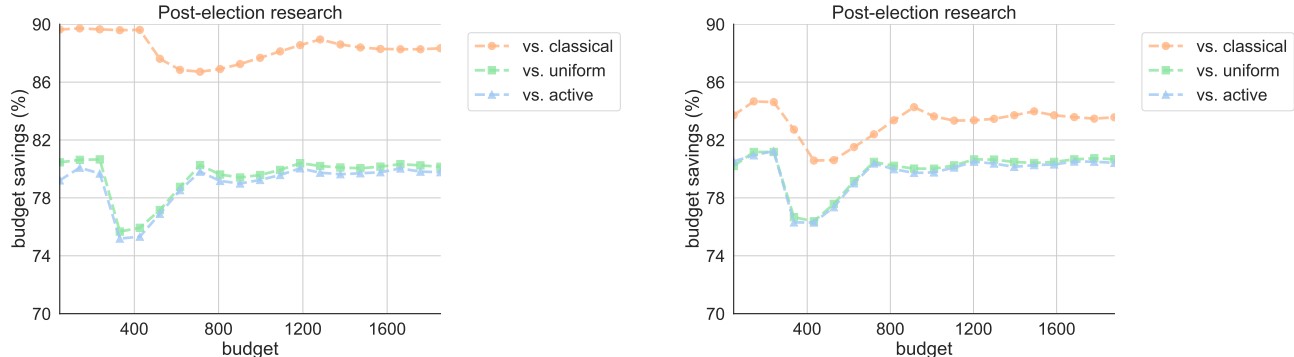

*Figure 6.* Percentage of budget saved by our method (relative to different baselines) when aiming to achieve the same performance at different budget levels.

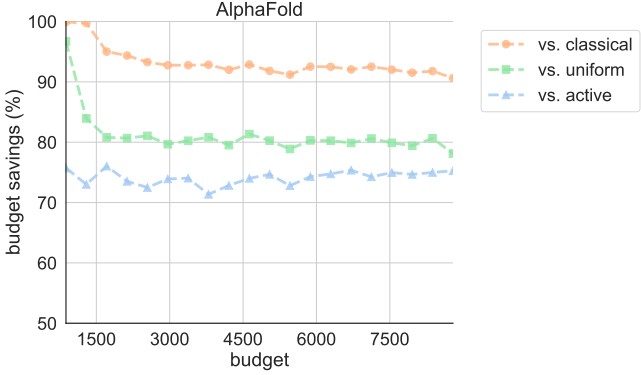

*Figure 7.* Percentage of budget saved, for the Alphafold experiment.

