# OpenReview forum: "Overcoming the Incentive Collapse Paradox"
_ICML.cc/2026/Conference — ICML 2026 regular_

### Official Review · Reviewer_K8Aw · 2026-02-17

**Soundness:** 2
**Presentation:** 3
**Significance:** 3
**Originality:** 3
**Overall Recommendation:** 4
**Confidence:** 3

**Summary:**

This paper focused on incentivizing human annotator efforts in AI-assisted labeling when the annotator effort is unobserved. It shows that accuracy-based payment schemes suffer from incentive collapse. When AI becomes more accurate, the annotator needs to be paid a great amount to invest their efforts. To avoid this, the authors propose sentinel-auditing: randomly inject a small fraction of “sentinel” tasks on which the AI should fail, and pay an additional bonus only when these sentinel tasks are answered correctly. This makes the marginal incentive for effort scale with the auditing rate rather than the error rate of AI. To allocate payments and audits among instances, the authors applied incentive-aware active statistical inference by optimizing sampling and auditing design under a single budget.

**Compliance With Llm Reviewing Policy:**

Affirmed.

**Final Justification:**

The authors' rebuttal solved my concerns, and I think the paper is novel and somewhat technically sound.

**Key Questions For Authors:**

How can you consider the strategic behavior of the annotators to reverse engineer the sentinel questions?

**Limitations:**

yes

**Strengths And Weaknesses:**

> Strengths

- The incentive collapse problem is a timely and important problem;
- The logic and reasoning of the paper are clear: the failure of accuracy-based payments makes sense to me, and the sentinel-auditing scheme is intuitive but effective.
- The statistical analysis is solild.

> Weaknesses

- My main concern is about the annotators themselves distinguishing the sentinel question or learn the patterns of the sentinel questions? I feel this might be possible to be modeled by strategic classification literature.

- The paper seems to assume the labeling is relatively easy to verify. If there are some difficult tasks and the ground truth answers are not available, it may be difficult to construct the sentinel question. The applicability may be limited.

---

> ### Author Rebuttal · Authors · 2026-03-31
>
> We thank the reviewer for their time and valuable feedback.
>
> **Response to W1 and Q1**
>
> We thank the reviewer for this helpful comment. We agree that an important practical issue is whether annotators can distinguish sentinel tasks or learn their patterns, and we will clarify this point in the revision.
>
> Regarding construction, we consider two ways to build sentinel tasks:
>
> (i) Deliberately editing the LLM response.
> We can deliberately edit the LLM response to create sentinel tasks. The advantage of this approach is that it does not affect the style of the LLM response, and it is relatively easy to generate in batch.
>
> (ii) Using the corrected history problem.
> Another approach is to use history tasks where the AI makes mistakes and humans correct them. The advantage is that this does not change the distribution of error types. The limitation is that these tasks depend on historical data, so they are harder to generate in batch.
>
> Based on these constructions, we believe it is generally hard to identify whether a task is a sentinel task **without checking the correctness of the answer itself**. In particular, under corrected-history construction this is especially difficult, while under deliberate editing it may be more possible if the induced errors have a different distribution.
>
> We also agree that this issue is related to the **strategic classification** literature [1][2]. Our framework can be extended to a setting where annotators form a belief about whether a task is sentinel. As long as they cannot **perfectly** identify sentinel tasks (i.e., their belief is strictly less than 1), the incentive effect can still be sustained. Suppose that for a non-sentinel task, the annotator believes it is sentinel with probability $\rho_{posterior}(X)$. Then, similarly to Theorem 4.1, the first-order condition becomes
> $$
> \rho_{posterior}(X) \mu(b(X)) q’(e(X)) = c'(e(X)),
> $$
>
> so a feasible effort level $e$ for non-sentinel tasks can still be supported.
>
> To further support this point, we added a sensitivity analysis in which $\rho_{posterior}(X)\equiv 0.8\rho$. The results are shown in https://anonymous.4open.science/w/Rebuttal-Material-for-Submission31103-4/ . As expected, the performance is somewhat worse than in the case where sentinel tasks are fully indistinguishable $(\rho_{posterior}(X)\equiv \rho)$, but our method still outperforms the baselines. We will add a discussion of this extension in the revision.
>
> **Response to W2**
>
> Thank you for this important comment. We agree that in practice the cost of verification or obtaining ground truth may differ substantially across tasks.
>
> Our framework can be extended to this case in a relatively simple way by classifying tasks by difficulty, say from $1$ to $H$. For each task $X_i$, let $h(X_i)$ denote its difficulty level. The probability that the annotator detects and corrects the error conditional on AI making mistake, $q$, depend not only on effort $e$ but also on $h(X_i)$, i.e., $q=q(e(X_i),h(X_i))$.
> In this case, the first-order condition in Theorem 4.1 still holds accordingly. Thus we can choose different auditing probabilities $\rho=\rho(h(X_i))$ or adjust the bonus $b(X_i,h(X_i))$, and therefore providing higher rewards and inducing higher effort on more difficult tasks. After replacing $\rho$, $q$, $b$, and $e$ accordingly, one can show that the estimator in Equation (5) remains unbiased.
>
> At the same time, the cost of obtaining ground truth or constructing sentinel tasks can also depend on difficulty, represented by $k(h(X_i))$. This changes the budget constraint to
> $$
> \mathbb E\left[\sum_{i=1}^n \Big(\rho(h(X_i)) b(X_i, h(X_i)) q(e(X_i), h(X_i))\xi_i + w_0 \xi_i+ \rho(h(X_i)) k(h(X_i))\Big) \right] \le B.
> $$
>
> To solve the above optimization problem, one possible approach is to decompose it into $H$ subproblems, each corresponding to a difficulty level. The budget allocation then proceeds in two stages: in the first stage, the total budget is allocated across difficulty levels; in the second stage, we apply the results from the current version of the paper (for homogeneous difficulty) to optimize the mechanism within each difficulty class.
>
>
> References
>
> [1] Hardt, Moritz, et al. "Strategic classification." Proceedings of the 2016 ACM Conference on Innovations in Theoretical Computer Science. 2016.
>
> [2] Ghalme, Ganesh, et al. "Strategic classification in the dark." International Conference on Machine Learning. 2021.

---

> > ### Author Rebuttal · Reviewer_K8Aw · 2026-03-31
> >
> > Thanks for the rebuttal. I am satisfied with the answers.

---

> > > ### Author Response · Authors · 2026-04-08
> > >
> > > We are very glad to hear that our response has resolved your concerns. Thank you again for your valuable comments and thoughtful feedback.

---

### Official Review · Reviewer_z6UD · 2026-02-26

**Soundness:** 3
**Presentation:** 3
**Significance:** 4
**Originality:** 4
**Overall Recommendation:** 5
**Confidence:** 4

**Summary:**

This paper considers the problem of incentivizing human annotators to provide AI-assisted labels. The problem setup is such that an AI helper labels a task correctly with probability $1 - p$, and a human worker corrects the AI’s label with a probability that scales with an “effort” parameter.

The work establishes that any reasonable accuracy-based payment scheme (payment increases with accuracy) must grow to infinity as the AI’s accuracy grows to one. A “sentinel-auditing” mechanism is then proposed, where an AI is deliberately forced to make errors on a subset of tasks. When a worker correctly answers the sentinel task, they are awarded a bonus payment. Under this setup, a strictly positive level of human effort can be guaranteed at finite cost, independent of the AI’s accuracy.

Section 5 then extends this to active statistical inference, where the mean of real-valued labels is to be estimated by actively querying for human input with probability $\pi$ that depends on the problem instance. An estimator is proposed and shown to be asymptotically normal. The asymptotic variance is then minimized with respect to all design choices (sentinel-auditing rate, bonus, $\pi$).

Semi-synthetic experiments on survey and protein data demonstrate improved cost-error tradeoffs compared to baselines that do not account for endogenous labeling costs.

**Compliance With Llm Reviewing Policy:**

Affirmed.

**Key Questions For Authors:**

The theoretical guarantees rely on the assumption that sentinel tasks are indistinguishable from natural AI errors. As $p \to 0$, natural errors may concentrate on “hard” or out-of-distribution examples. A rational annotator may then be able to detect sentinels with low effort, and may game the system and exert effort only on the sentinels. Does Theorem 4.1 hold if the annotator’s effort $e$ is conditional on the posterior probability of a task being a sentinel?

**Limitations:**

Yes.

**Strengths And Weaknesses:**

**Soundness**

- Strength: The theoretical formalization of the incentive collapse (Theorem 3.1) and the sentinel solution (Theorem 4.1) is rigorous and intuitive. The conclusion that accuracy-based payments must scale with the inverse of the AI error rate shows that naive labeling paradigms will fail as AI assistants improve.

- Weakness: The incentive-robust payments critically relies on the ability of the requestor to construct sentinel tasks. Designing sentinel tasks that ensure an AI will make errors can be difficult, and infeasible when the fraction of sentinels $\rho$ is large.

- Weakness: Section 5 considers the setting of continuous-valued labels ($Y \in \mathbb{R}$). A more reasonable model of worker behavior may be to assume that a worker can catch a sentinel, but may provide a noisy correction. .

- Weakness: The model assumes that workers are reliable, in that if an AI submits a correct label, the worker will not mistakenly introduce errors.

**Presentation**

- Strength: The paper is logically structured, moving from an impossibility result to mechanism design and statistical inference.

* Weakness: The framing of Theorem 3.1 as a “paradox” is hyperbolic. The fact that the incentive scales grows to infinity as $p \to 0$ is a reasonable consequence of rare event detection, rather than a counter-intuitive economic phenomenon. The paper treats this as a major impossibility result, when it is arguably a straightforward application of the cost of monitoring rare events.


**Significance**

- Strength: The work addresses a timely and critical problem. As large language models become highly accurate, crowdsourced annotations are often likely to be labeled by LLMs and verified/corrected by human annotators. The paper convincingly argues that naive mechanisms do not scale economically.


**Originality**

- Strength: The joint optimization framework in Section 5, which solves for the sampling probability $\pi$, auditing rate $\rho$, and bonus $b$, is an original contribution to active inference.

---

> ### Author Rebuttal · Authors · 2026-03-31
>
> We thank the reviewer for their time and valuable feedback. We also appreciate the reviewer’s acknowledgment of the strengths of our work.
>
> **Response to Soundness W1**
>
> Thank you for this helpful comment. We agree that constructing sentinel tasks is an important practical challenge. Besides designing sentinel tasks on which the AI is likely to fail, one can **force** the AI suggestion shown to the annotator to be incorrect, for example by deliberately modifying the AI's original answer before presenting it.
>
> More specifically, we consider the following two ways to construct sentinel tasks:
>
> (i) Deliberately editing the LLM response.
> We can deliberately edit the LLM response to create sentinel tasks. The advantage of this approach is that it does not affect the style of the LLM response, and it is relatively easy to generate in batch.
>
> (ii) Using the corrected history problem.
> Another approach is to use history tasks where the AI makes mistakes and humans correct them. The advantage is that this does not change the distribution of error types. The limitation is that these tasks depend on historical data, so they are harder to generate in batch.
>
> **Response to Q1**
>
> Thank you for this important comment. This is indeed an important practical issue, and the answer is yes: Theorem 4.1 can be extended to this setting. Although, as noted in our response to Soundness W1, our framework does not necessarily rely on natural AI errors, it is still possible that annotators can sometimes recognize sentinel tasks, for example from the error distribution as you suggested.
>
> Our framework can allow annotators to form a belief about whether a task is sentinel. As long as they cannot **perfectly** identify sentinel tasks (i.e., their belief is strictly less than 1), the incentive effect can still be sustained. Suppose that for a non-sentinel task, the annotator believes it is sentinel with probability $\rho_{posterior}(X)$. Then, similarly to Theorem 4.1, the first-order condition becomes
> $$
> \rho_{posterior}(X) \mu(b(X)) q’(e(X)) = c'(e(X)),
> $$
>
> so a feasible effort level $e$ for non-sentinel tasks can still be supported.
>
> To further support this point, we added a sensitivity analysis in which $\rho_{posterior}(X)\equiv 0.8\rho$. The results are shown in https://anonymous.4open.science/w/Rebuttal-Material-for-Submission31103-4/ . As expected, the performance is somewhat worse than in the case where sentinel tasks are fully indistinguishable $(\rho_{posterior}(X)\equiv \rho)$, but our method still outperforms the baselines. We will add a discussion of this extension in the revision.
>
> **Response to Soundness W2**
>
> Thank you for this helpful suggestion. We agree that a more realistic model is that a worker may provide a **noisy correction**.
>
> Our framework can be extended to handle this case by making the bonus depend on the **quality of the correction**. Concretely, suppose that after detecting the error, the worker produces a correction with quality parameter $\phi \in [0,1]$, where $\phi$ follows some distribution, and the payment depends on $b\phi$. Then the first-order condition in Theorem 4.1 is modified to
>
> $$
> \rho \mathbb{E}[\mu(\\phi b(X))] q'(e^\star (X))= c'(e^\star (X)).
> $$
>
> We will add a discussion of this extension in the revision.
>
> **Response to Soundness W3**
>
> Thank you for this important comment. We agree that workers may introduce mistakes when the AI label is already correct. However, our framework can be extended to allow this type of worker error.
>
> Specifically, suppose the worker makes a mistaken change when the AI label is correct with probability $p_{\mathrm{mis}}=\psi p$. Since this only happens on non-sentinel tasks (the AI output on sentinel tasks is deliberately forced to be incorrect), it does not affect the payment rule and therefore does not affect the worker’s effort choice. The main change is that we need to correct for the bias this introduces in the estimator in Equation (5). In this case, we can modify the estimator to
> $$
> \hat{\theta}^{\pi}
> =\frac{1}{n}\sum_{i=1}^n \left[
> f(X_i)+(Y_i - f(X_i))
> \frac{\xi_i\zeta_i/(1-\rho)}{\pi(X_i)(q(e(X_i))-(1-p)\psi)}
> \right].
> $$
>
> Similarly, one can show that this modified estimator is unbiased. In practice, especially when $p$ is small, the term $(1-p)$ may be approximated by $1$.
>
>
> **Response to Presentation W1**
>
> Thank you for this comment. We agree that the wording here may be too strong, and we will revise it in the paper.

---

> > ### Author Rebuttal · Reviewer_z6UD · 2026-04-01
> >
> > Thank you for the response. I have read the responses to all reviewers, and most of my questions have been addressed.
> >
> > However, I am a bit skeptical regarding the proposed design of sentinel tasks. Method (ii) works, but it is indeed harder to scale. Method (i) works when the task is designed as a question regarding categorical labels. However, it does not seem to work for continuous labels (which is within the scope of this manuscript) as described in the following example.
> >
> > Consider for example the task of estimating the concentration level of a protein $Y \in \mathbb{R}$ in milligrams [mg], and the AI predicts $y = 120$. According to method (i), the response $y$ can be edited to $y' = 118$. An annotator exerting high effort may, independent of the AI's output, calculate $y^* = 119$ and decide that $y'$ is within an acceptable margin of error. The proposed mechanism registers this as zero effort, and heavily penalizes the high-effort worker.
> >
> > If the edit's magnitude is larger, e.g., $y \mapsto y' = 10$, then the error may be obvious at a glance. The annotator no longer needs to exert high effort, and the sentinel can easily be spotted. The annotator can claim the bonus $b(x)$ without much difficulty.
> >
> > Despite these challenges regarding sentinel construction, a concern shared by other reviewers, I believe the paper's core theoretical formulation of the incentive collapse and its integration with active inference remains highly valuable. Therefore, I will maintain my current positive score, but I invite the authors to briefly address the specific margin-of-error tradeoff for real-valued tasks in their final response or acknowledge it as a limitation in the final manuscript.

---

> > > ### Author Response · Authors · 2026-04-01
> > >
> > > Thank you for this thoughtful follow-up and for your positive assessment of our paper. We also appreciate your clarification of the concern for continuous-label tasks.
> > >
> > >
> > > We agree that, in the continuous setting, choosing the magnitude of the deliberate modification is important. There is indeed a key tradeoff here between the acceptable margin of error and the need to keep the induced error distribution realistic.
> > >
> > >
> > > **We think this issue can be viewed in two cases.**
> > >
> > > **Case 1:** The cost of checking carefully is high, but the cost of correction is relatively low.
> > > Examples include proofreading or polishing AI-generated text [1]; refining AI-generated bounding boxes or correcting AI-generated object/region boundaries in image annotation tasks [2]; medical-imaging workflows such as longitudinal CT follow-up, where baseline and follow-up scans are first aligned so that the system can localize the lesion, and the AI model then proposes a lesion contour or volumetric segmentation for the radiologist to confirm or edit [3][4][5]. In such settings, this is less problematic: the error distribution of sentinel tasks does not need to exactly match the natural AI error distribution, as long as the sentinel tasks keep annotators attentive and willing to verify the AI output carefully.
> > >
> > >
> > > **Case 2:** The correction cost is itself non-negligible. In this setting, the problem is more challenging. One possible approach is to consider the setting, as suggested by the reviewer and discussed in our rebuttal, where a strategic reviewer can form posterior beliefs and potentially recognize sentinel tasks. However, we agree that this case is substantially more difficult in practice, and deliberately modifying the AI response in a way that is both effective and natural may require considerable care. We acknowledge this as a limitation of the current paper and view it as an important direction for future work.
> > >
> > >
> > > At the same time, as the reviewer noted, we believe our method still applies to many important scenarios, such as classification tasks or settings where the correction cost is low. We thank the reviewer again for highlighting this point, and we will clarify this limitation and scope more explicitly in the final manuscript.
> > >
> > >
> > > References
> > >
> > > [1] Artemova, Ekaterina, et al. "Beemo: Benchmark of expert-edited machine-generated outputs." Proceedings of the 2025 conference of the Nations of the Americas Chapter of the Association for Computational Linguistics, 2025.
> > >
> > > [2] Cormier, Mickael, et al. "Interactive labeling for human pose estimation in surveillance videos." Proceedings of the IEEE/CVF international conference on computer vision. 2021.
> > >
> > > [3] Hering, Alessa, et al. "Improving assessment of lesions in longitudinal CT scans: a bi-institutional reader study on an AI-assisted registration and volumetric segmentation workflow." International journal of computer assisted radiology and surgery. 2024
> > >
> > > [4] Murugesan, Gowtham Krishnan, et al. "AI generated annotations for Breast, Brain, Liver, Lungs, and Prostate cancer collections in the National Cancer Institute Imaging Data Commons." Scientific Data, 2025.
> > >
> > > [5] Jabbour, Sarah, et al. "Measuring the impact of AI in the diagnosis of hospitalized patients: a randomized clinical vignette survey study." JAMA, 2023.

---

### Official Review · Reviewer_5gNq · 2026-03-12

**Soundness:** 3
**Presentation:** 3
**Significance:** 3
**Originality:** 3
**Overall Recommendation:** 4
**Confidence:** 3

**Summary:**

This paper studies the Awakening Crowd of Experts (ACE) problem, where new experts arrive each round and remain available. In the stochastic setting, a pessimistic lower-confidence algorithm is proposed to avoid overestimating new experts, achieving near-optimal regret with matching lower bounds. In the adversarial setting, sublinear α-regret is impossible for constant α; the authors design an exploration–exploitation algorithm that balances established and newly arrived experts, achieving near-optimal performance. This work provides the first regret analysis for online learning with a growing expert pool, offering both theoretical guarantees and tailored algorithms.

**Compliance With Llm Reviewing Policy:**

Affirmed.

**Key Questions For Authors:**

Please refer to the “Weaknesses”

**Limitations:**

Please refer to the “Weaknesses”

**Strengths And Weaknesses:**

Strengths
1. Strong Theory
Shows why traditional payment plans fail as AI improves (Theorem 3.1).
Sentinel-Auditing guarantees high-quality human work at fixed cost (Theorem 4.1).
Data is statistically valid (consistency and asymptotic normality).
Framework balances budget, auditing, and sampling for accuracy.
2. Clear Writing
Logical flow: problem → cause → solution → usage.
Intuitive explanations before math.
Concrete examples illustrate theory.
Abstract/intro clearly highlight main contributions.

Weaknesses
1 Idealization of Sentinel Tasks
The paper’s mechanism relies on sentinel tasks to incentivize annotator effort. However, it seems idealized: the AI is assumed to fail on these tasks. In practice, the AI may occasionally produce correct labels even on sentinel tasks. The impact of such occasional correct AI outputs on the strength of incentives and the robustness of the mechanism is not discussed, leaving a gap in understanding its behavior in realistic settings.
2 Unclear Assumptions in Experiments
It is not explicitly stated whether the experiments assume that the AI always fails on sentinel tasks. Clarifying this assumption would help evaluate the applicability of the proposed mechanism in real-world scenarios where AI errors are stochastic.
3 Fixed Auditing Probability
The auditing probability  𝜌 is fixed across tasks in the proposed mechanism. The potential benefits of dynamically adjusting  ρ (e.g., increasing auditing when sentinel tasks are frequently answered incorrectly) are not considered. A discussion of adaptive auditing could help clarify practical robustness and efficiency in budget allocation.

---

> ### Author Rebuttal · Authors · 2026-03-31
>
> We thank the reviewer for their time and valuable feedback.
>
> **Response to W1**
>
> Thank you for this important comment. Our sentinel tasks do  **not necessarily**, rely on the AI naturally producing wrong answers on those tasks. Instead, the key idea is that we can **force** the AI suggestion shown to the annotator to be incorrect, for example by deliberately modifying the AI's original answer before presenting it. More specifically, we can consider the following construction methods:
>
> (i) Deliberately editing the LLM response.
> We can deliberately edit the LLM response to create sentinel tasks. The advantage of this approach is that it does not affect the style of the LLM response, and it is relatively easy to generate in batch.
>
> (ii) Using the corrected history problem.
> Another approach is to use history tasks where the AI makes mistakes and humans correct them. The advantage is that this does not change the distribution of error types. The limitation is that these tasks depend on historical data, so they are harder to generate in batch.
>
> We will clarify this point explicitly in the revised version.
>
> **Response to W2**
>
> Thank you for this comment. In our experiments, we directly modify the AI provided answer on sentinel tasks and therefore we can **force** the AI suggestion to be incorrect, rather than assuming that the AI naturally fails on those tasks. We will clarify this point explicitly in the revised version.
>
> **Response to W3**
>
> Thank you for this helpful comment. We agree that allowing the auditing probability to vary across tasks is important in practice. Intuitively, $\rho$ should depend on task difficulty, since harder tasks may require a higher auditing probability to provide stronger incentives for annotators.
>
> In fact, our framework can be extended in a relatively straightforward way by replacing the fixed $\rho$ with a task-dependent $\rho(X)$. For example, Equation (5) can be modified by replacing $\rho$ with $\rho(X_i)$, and similarly the objective in Lemma 5.2 becomes minimizing
> $$
> \mathbb E \left[\frac{(Y^{\text{true}}-f(X))^2}{(1-\rho(X))\pi(X)q(e(X))}\right]
> $$
> over $(\pi,\rho,b)$. The main challenge is that this makes the corresponding optimization problem substantially more difficult, and one may also need to explicitly model or estimate task difficulty.
>
> Specifically, we may assume we can classify tasks by difficulty, say from $1$ to $H$. For each task $X_i$, let $h(X_i)$ denote its difficulty level. The probability that the annotator detects and corrects the error conditional on AI making mistake, $q$, depend not only on effort $e$ but also on $h(X_i)$, i.e., $q=q(e(X_i),h(X_i))$.
> In this case, the first-order condition in Theorem 4.1 still holds accordingly. So our method still guarantees non-vanishing human effort.Moreover, we can choose different auditing probabilities $\rho=\rho(h(X_i))$ or adjust the bonus $b(X_i,h(X_i))$, and therefore providing higher rewards and inducing higher effort on more difficult tasks. After replacing $\rho$, $q$, $b$, and $e$ accordingly, one can show that the estimator in Equation (5) remains unbiased.
>
> At the same time, the cost of obtaining ground truth or constructing sentinel tasks can also depend on difficulty, represented by $k(h(X_i))$. This changes the budget constraint to
> $$
> \mathbb E\left[\sum_{i=1}^n \Big(\rho(h(X_i)) b(X_i, h(X_i)) q(e(X_i), h(X_i))\xi_i + w_0 \xi_i+ \rho(h(X_i)) k(h(X_i))\Big) \right] \le B.
> $$
>
> To solve the above optimization problem, one possible approach is to decompose it into $H$ subproblems, each corresponding to a difficulty level. The budget allocation then proceeds in two stages: in the first stage, the total budget is allocated across difficulty levels; in the second stage, we apply the results from the current version of the paper (for homogeneous difficulty) to optimize the mechanism within each difficulty class.
>
> In addition, one could also consider adaptively updating the auditing probability online, as the reviewer suggested, for example by increasing $\rho$ for task types on which annotators frequently fail sentinel tasks. Such an adaptive design may further improve robustness in practice.
>
> We will add a discussion of this possible extension in the revised version.

---

### Official Review · Reviewer_sxaj · 2026-03-13

**Soundness:** 2
**Presentation:** 3
**Significance:** 3
**Originality:** 3
**Overall Recommendation:** 3
**Confidence:** 3

**Summary:**

This paper studies a core challenge in AI-assisted labeling. When AI systems become more accurate, standard payment schemes based on labeling accuracy won't work. The reason is that strategic human annotators can just free-ride using the strong AI. But preventing this requires payments that grow unboundedly as AI error rates decrease. This is termed the incentive collapse paradox.

The paper proposes sentinel auditing as one way to fix this, where in sentinel tasks the correctness can be checked. Then based on this, the paper goes further into adapting statistical inference to account for the strategic behavior of annotators. The proposed idea is to jointly optimizes how often to inject sentinel tasks, which instances to query for labels, and how much extra payment to offer per instance. This joint design leads to better confidence intervals and better cost-accuracy tradeoffs. Experiments on two datasets from different domains show the effectiveness of the method.

**Compliance With Llm Reviewing Policy:**

Affirmed.

**Key Questions For Authors:**

Here are my main questions:

1a) So I think the whole framework is predicated on there being a sentinel task.
It would be great if the authors could offer more justification on when/how these may be constructed.

1b) Moreover, the strategic annotators are somehow not able to tell if it is labeling for a sentinel task or not.
In this respect, I find the paper's modeling less convincing.

2) The authors make strong assumptions on knowing various aspects of the annotator's best response.
This includes the cost function and the utility function.
I would like to see more justification for this as ostensibly it is a bit strong.

To add, the usual way this assumption can be relaxed is via online learning, in face of unknown annotator optimization parameters.
I wonder if such a result may be possible.

3)  It seems that the theory results in the paper use that Y is a real value scalar.
But in the two experiments, Y is discrete and binary.
This makes me wonder if there is an experiment that can illustrate the method's effectiveness to handle continuous Y?

4) Can the authors mention the specific labeling settings where their framework is most applicable?
For instance, it is not immediately clear how the framework applies to modern LLM training pipelines, where human effort is mainly on building evaluation environments rather than instance-level labeling.
But of course, this is only one such training task.
What are the training tasks the authors are thinking of that would be best addressed by the methods in the paper?

**Limitations:**

The paper doesn't seem to have a limitation subsection, but there are some comments on it.
It may be good to have a separate subsection for this in the interest of transparency.

**Strengths And Weaknesses:**

Soundness:
The paper has ample theoretical results.
I thought Theorem 3.1 was pretty interesting and offers a nice motivation for the problem as it is framed.
I think the sentinel solution from Section 4 is predicated on there being a sentinel task (more questions on this later).


Presentation:
The paper is clear and well written for the most part.
The paper would do well to further tighten the notation.
For instance, in section 2, when Y is introduced, please say if it is discrete or a real valued scalar.

Significance:
I think the problem studied is interesting if one can be convinced of the premise set forth by the paper.
I do think the paper's approach does require knowledge of various aspects of the annotator optimization (best response).
For example, it is assumed that annotator cost is known and the utility function are known.
Moreover, besides being able to devise sentinel tasks, it is assumed that the annotators themselves cannot tell if the task is sentinel.

Originality:
To my knowledge, this work does yield original results and combining mechanism design with active statistical inference (Section 5).
Because mechanism design for data labeling is so dense, I do wonder if there are connections to other pieces of existing work.
It would be great if another reviewer more familiar with this could chime in on this as I'm not an expert on this topic.

---

> ### Author Rebuttal · Authors · 2026-03-31
>
> We thank the reviewer for their time and valuable feedback.
>
> **Response to Q1**
>
> Thank you for raising this important point. We agree that the practicality of our framework depends on how sentinel tasks are constructed and whether annotators can identify them. We will clarify both issues and add a discussion in the revision.
>
> 1a) We consider two possible ways to construct sentinel tasks.
>
> (i) Deliberately editing the LLM response.
> We can deliberately edit the LLM response to create sentinel tasks. The advantage of this approach is that it does not affect the style of the LLM response, and it is relatively easy to generate in batch.
>
> (ii) Using the corrected history problem.
> Another approach is to use history tasks where the AI makes mistakes and humans correct them. The advantage is that this does not change the distribution of error types. The limitation is that these tasks depend on historical data, so they are harder to generate in batch.
>
> 1b) What if annotators can identify sentinel tasks?
>
> This is an important issue. It is possible that annotators can sometimes recognize sentinel tasks. For the two constructions above, we believe the situation is different: Under **(ii)**, it is hard to tell; under **(i)**, it may be possible because the error distribution can look different.
>
> However, our framework can be extended to this case as long as annotators cannot **perfectly** classify whether a task is sentinel. Suppose that for a non-sentinel task, the annotator believes it is sentinel with probability $\rho_{posterior}(X)$. Then, similarly to Theorem 4.1, the first-order condition becomes$$\rho_{posterior}(X) \mu(b(X)) q’(e(X)) = c'(e(X)),$$
> so a feasible effort level $e$ for non-sentinel tasks can still be supported.
>
> **Response to Q2**
>
> Thank you for this comment. Our theory indeed depends on the specification of the utility and cost functions. In practice, one can use prior data to learn about human annotator behavior, or adaptively learn it over time (e.g., via online learning), as you suggested.
>
> We also note that even with misspecified utility and cost functions, our method still guarantees non-vanishing human effort and outperforms baselines that ignore strategic behavior. We include a new sensitivity analysis experiment to support this claim. The results are shown in https://anonymous.4open.science/w/Rebuttal-Material-for-Submission31103-1/ . Consistent with the findings in our paper, our method provides better confidence intervals than the baselines. We will add a discussion in the revision.
>
> **Response to Q3**
>
> Thank you for the comment. We have added a new experiment in a continuous setting using the 2019 ACS PUMS data for California [1][2], where $Y$ is income. The ACS PUMS is a large-scale annual survey conducted by the U.S. Census Bureau that contains demographic information such as age, sex, education, and income. In this experiment, our goal is to estimate the linear regression coefficient of age when regressing income on age and sex, using the general M-estimation framework in Section 5.3. The results are shown in https://anonymous.4open.science/w/Rebuttal-Material-for-Submission31103-2/ . Our method achieves better confidence intervals than the baselines.
>
> **Response to Q4**
>
> Thanks for this awesome question. We believe our method can be useful in a wide range of human-AI collaboration settings, including both domain science applications and modern LLM training pipelines. In domain science, AI is increasingly used to assist human decision making in areas such as child welfare [3], wildlife conservation [4], and healthcare [5]. For example, in healthcare, clinicians may diagnose patients based on information from chest X-rays and patient history, often with the assistance of AI tools [5]. At the same time, researchers may wish to collect labeled data to better understand disease mechanisms. In such settings, clinicians can be paid to provide labels. However, if clinicians have access to AI predictions, their labels may be influenced by the AI system, leading to systematic bias in the collected data. In modern LLM training pipelines, our framework is also applicable. Since our approach is based on *general M-estimation*, it naturally covers many loss functions used in practice, including those arising in *reward model training*.
>
> **Limitation**
>
> Thanks for your comment. We will add a separate **Limitations** section in the revision.
>
> References
>
> [1] U.S. Census Bureau, American Community Survey (ACS) Public Use Microdata Sample (PUMS), 2019.
>
> [2] Ding, et al. "Retiring adult: New datasets for fair machine learning." NeurIPS, 2021.
>
> [3] De-Arteaga, et al. "A case for humans-in-the-loop: Decisions in the presence of erroneous algorithmic scores." CHI, 2020.
>
> [4] Beery, et al. "Efficient pipeline for camera trap image review." arXiv, 2019.
>
> [5] Jabbour, et al. "Measuring the impact of AI in the diagnosis of hospitalized patients: a randomized clinical vignette survey study." JAMA, 2023.

---

> > ### Author Rebuttal · Reviewer_sxaj · 2026-04-02
> >
> > I thank the authors for writing up their rebuttal. On the balance, I do still have some lingering questions about the paper's overall setup. Chiefly, since annotators have access to AI, they can also just use the AI to detect sentinel tasks. And for justifying settings where this is relevant, I guess one thought that couldn't this be more of a logistical issue? One idea is just for clinicians to have them come in and give the diagnosis. We can be confident they don't use AI simply because on the computer they are viewing the case, we disable access to AI. For medical professionals, it also seem like a violation of their ethical code to be giving false labels. Regardless, I'm not sure about the specifics of data labeling myself and it is entirely possible clinicians do give false labels (or maybe it's just human error). Overall, it just would be good to see some concrete settings where this type of setup makes the most sense.

---

> > > ### Author Response · Authors · 2026-04-05
> > >
> > > Thank you for this thoughtful follow-up and for engaging so carefully with the overall setup of the paper. We appreciate this concern, and we agree that the practical relevance of the setup should be clarified more concretely.
> > >
> > > **[Regarding the concern that annotators may use AI to detect sentinel tasks]**
> > >
> > > We think there are two cases.
> > >
> > > Case 1: Sentinel tasks based on genuine AI errors.
> > >  This corresponds to method (ii) in response to W1, i.e., corrected-history tasks. In this case, the displayed AI output is a real historical AI mistake, there is no obvious pattern for annotators to detect by consulting AI.
> > >
> > > Case 2: Sentinel tasks based on deliberately edited AI outputs.
> > >  This corresponds to method (i) in response to W1. In principle, an annotator might try to use external AI tools to detect whether a task is a sentinel. However, in real labeling workflows, annotation is performed on dedicated platforms and interfaces, where AI-generated pre-labels are presented directly in the annotation UI for annotators to review or correct [1,2,3,4]. For example, Labelbox provides built-in workflows and detailed guidance for model-assisted labeling, where model predictions can be imported directly into the annotation editor and then reviewed by human annotators in the UI [1]. Other widely used labeling platforms, such as Label Studio [2], CVAT [3], and Encord [4], also support similar workflows.
> > >
> > > In such settings, consulting an external AI system is often inconvenient, as it incurs additional time and cost, and may also violate workflow rules due to data privacy or project-specific requirements [5]. Moreover, in some deployments, the use of external AI tools can simply be disallowed as part of the labeling protocol. For this reason, we believe that using AI to detect sentinel tasks is unlikely to be a major issue in practice.
> > >
> > > **[Why not simply ban AI use?]**
> > >
> > > In simple terms, directly banning all the use of AI through logistical restrictions is often undesirable, because AI suggestions can substantially improve labeling efficiency. A direct indication that this setup is practically relevant is that many labeling platforms already provide model-assisted labeling directly inside the annotation UI [1,2,3,4]. Commercial AI data companies such as Scale AI also provide Human-AI workflows for generating data labels [6].
> > >
> > > **[Concrete settings]**
> > >
> > > One concrete setting where we believe our method is especially relevant is image annotation, such as refining AI-generated bounding boxes, correcting AI-generated object or region boundaries, or verifying predicted object categories [7]. Such data are widely needed for downstream machine learning applications, including autonomous driving, medical imaging, surveillance, retail, agriculture, and wildlife monitoring. For example, a principal may wish to label objects in images in order to train an object detection model. In such settings, AI can first provide an initial detection or annotation, and the human annotator only needs to check the AI output and make corrections when necessary. This can greatly improve labeling efficiency, but it also creates the risk that annotators rely too heavily on the AI output and fail to verify it carefully. This is a common image-labeling workflow, as illustrated by the documentation of major labeling platforms [1,2,3,4].
> > >
> > > Other realistic settings include proofreading or polishing AI-generated text [8], as well as video annotation and tracking workflows [3]. We also apologize for any confusion caused by our healthcare example. What we mainly had in mind was dataset annotation for model development, where clinicians or radiologists are recruited to provide labels for research data [9], rather than real-time patient care.
> > >
> > > To summarize, we agree that the setup is more natural in some settings than in others, and we will clarify this more carefully in the revision. At the same time, we believe this is a practical setting with many real applications. We hope this clarification addresses the reviewer’s concern, and we thank the reviewer again for pushing us to make the practical scope of the paper more explicit.
> > >
> > > [1] Labelbox, “Import annotations as pre-labels”, documentation, accessed 2026.4.2
> > >
> > > [2] Label Studio, ”Machine learning”, documentation, accessed 2026.4.2
> > >
> > > [3] CVAT, “Annotation”, documentation, accessed 2026.4.2
> > >
> > > [4] Encord, “Modalities”, accessed 2026.4.2
> > >
> > > [5] CVAT, “Lecture 3. Data Confidentiality in Annotation: Rules, Risks, and Best Practices”, accessed 2026.4.2
> > >
> > > [6] Wang, et al. “Prelabeling for semantic segmentation tasks.” U.S. Patent, 2023
> > >
> > > [7] Cormier, et al. "Interactive labeling for human pose estimation in surveillance videos."ICCV Workshop, 2021.
> > >
> > > [8] Artemova, et al. "Beemo: Benchmark of expert-edited machine-generated outputs."NAACL, 2025.
> > >
> > > [9] Murugesan, et al. "AI generated annotations for Breast, Brain, Liver, Lungs, and Prostate cancer collections in the National Cancer Institute Imaging Data Commons." Scientific Data, 2025.

---

### Decision · Program_Chairs · 2026-04-30

**Decision:**

Accept (regular)

**Comment:**

The reviewers acknowledge that this paper studies a timely and important problem, and the sentinel solution is rigorous and intuitive. During the discussion phase the authors addressed the reviewers' questions about heterogeneity of the task difficulty, annotator has some ability in detecting sentinel tasks, noisy correction, etc, with experimental support. There were still some concerns about the real-world motivations and the strong assumptions, but overall the merits outweight the concerns.